# Computational analysis of stochastic delay dynamics in maize streak virus

**Sana Iqbal[1], Naveed Shahid[1], Ali Raza[2,3]\*, Marek Lampart[3,4], Nauman Ahmed[1,5], Dumitru Baleanu[6], Hala H. Taha[7]**

**1** Department of Mathematics and Statistics, The University of Lahore, Lahore, Pakistan, **2** Department of Mathematics, Center for Research and Development in Mathematics and Application (CIDMA), University of Aveiro, Aveiro, Portugal, **3** IT4Innovations, VSB-Technical University of Ostrava, Ostrava, Czech Republic, **4** Department of Applied Mathematics, VSB Technical University of Ostrava, Ostrava, Czech Republic, **5** Jadara Research Center, Jadara University, Jordan, **6** Department of Computer Science and Mathematics, Lebanese American University, Beirut, Lebanon, **7** Department of Mathematical Sciences, College of Science, Princess Nourah bint Abdulrahman University, Riyadh, Saudi Arabia

\* ali.raza@vsb.cz (AR)

## Abstract

### Objectives

The primary goal of this research is to analyze the transmission dynamics of Maize Streak Virus (MSV) by means of a computational and stochastic modeling technique where the time delay and uncertainty factors in the epidemic process are vital considerations.

### Methodology

A compartmental MSV deterministic model was established, which later got an extension to a stochastic delay differential system having five biological compartments consisting of susceptible, insecticide-treated, exposed, infected, and recovered plants. Analytical methods were employed to find the maize streak–free and endemic equilibriums and to derive the treatment reproduction number. The stability of the deterministic and stochastic systems was studied. The numerical methods used for comparison were Euler-Maruyama, stochastic Runge–Kutta, and the stochastic Nonstandard Finite Difference (NSFD) scheme, which were assessed for accuracy, stability, and computational efficiency.

### Key Results

Theoretical results show that under some parameter values, both equilibrium points are stable in an asymptotic sense. The numerical experiments reveal that the stochastic NSFD scheme is more stable, preserves positivity better, and is independent of step size than the classical methods. Including the stochasticity captures the uncertainty associated with MSV transmission in the real world, thereby enhancing the predictive simulation's validity.

**Data availability statement:** All data underlying the findings of this study are fully available within the manuscript. No additional Supporting Information files or external datasets exist.

**Funding:** This article has been produced with the financial support of the European Union under the REFRESH – Research Excellence for REgion Sustainability and High-tech Industries project number CZ.10.03.01/00/22_003/0000048 via the Operational Programme Just Transition. One of the authors (D.B.) would like to thank for the grant funding PIRF Project Number: I0074 provided by the Lebanese American University, Beirut, Lebanon. Also, this work was supported by Princess Nourah bint Abdulrahman University Researchers Supporting Project number (PNURSP2025R899), Princess Nourah bint Abdulrahman University, Riyadh, Saudi Arabia.

**Competing interests:** The authors have declared that no competing interests exist.

## Conclusions

The suggested stochastic NSFD model is indeed a strong computationally efficient and biologically realistic method to simulate MSV and other plant virus epidemics. The results boost our understanding and management of the agricultural disease control strategies.

## 1 Introduction

Maize (Zea mays) or corn is one of the world's most important cereal crops, globally produced. Its domestication in Mesoamerica has rendered it a cornerstone of world agricultural production and a prime source of food security and economic development [1]. Maize production is threatened by Maize Streak Virus (MSV), a leafhopper-transmitted disease-causing significant loss in yield, particularly in the tropical and subtropical regions of the world. There have been attempts at modeling MSV transmission dynamics via various mathematical and computational approaches. For instance, Seidu [2] proposed a deterministic ODE model involving fractional-order derivatives—i.e., the Atangana–Baleanu Caputo-type operator to capture memory effects and non-local interactions more accurately than traditional approaches. Liu [3] proposed an integrated stochastic model of variability in infection dynamics due to random environmental factors, modeling infection fluctuations via a logarithmic Ornstein–Uhlenbeck process. Mrope and Kigodi [4] gave an elaborate review of MSV control and transmission models in agroecosystems while O'Halloran et al. [5] researched the implementation of advanced deep-learning techniques for early detection of maize disease. They worked on the basis of integrating artificial intelligence for real-time monitoring of disease to enhance the responsiveness and efficiency of agricultural health systems. In another study, Ackora-Prah et al. [6] examined disease interactions within maize farms with Holling's functional response within a fractal–fractional setting and showed that such models better capture biological complexities. We extend these efforts by developing a stochastic delayed model for which positivity and stability are assured. By the Newton polynomial routine, we carried out numerical simulations to examine the qualitative behavior of the model and confirm theoretical results. A few notable contributions are studies conducted by Mrope and Kigodi [7], in which they investigated the dynamic interaction of maize plants with Homopteran insect virus vectors. Facchi et al. [8] proposed the use of chitosan- and tannin-based polymeric coatings as antimicrobial agents for the management of Xanthomonas vasicola pv. vasculorum (Xvv), in which they demonstrated promising applications at the field level. Ali and Ameen [9] applied fractional calculus to investigate MSV persistence and transmission and noted its application in developing disease control policies. Dash and Sethy [10] noted that maize infections are a major cause of production loss but can be avoided by early detection and prevention. Kalyango and Ntanda [11] created an explainable deep-learning model for the diagnosis of maize diseases with the trade-off between predictive performance and explainability in order to facilitate effective agricultural decision-making. Suriani et al.

[12] documented in morphological, physiological, and molecular detail the pathogens of bacterial stalk rot in maize, making possible species-level diagnosis. Malar et al. [13] applied Caputo–Fabrizio fractional derivative to describe MSV complex dynamics, while Mrope and Kigodi [14] also took into account the effects of control actions that are insecticide-based in an effort to decrease infection levels. Other deterministic and fractional models [15,16] have also provided information about the persistence of infection, memory effects, and long-term maize epidemic dynamics.

Beyond MSV-specific studies, more general stochastic epidemic models incorporating mechanical, chemical, and preventative control measures [17] have highlighted the central position of multi-strategy methods in reducing infection prevalence. Robaina et al. [18] standardized the inoculation protocol of Xvv in maize and determined a diagrammatic scale for resistance screening, and Tembo et al. [19] reported a quick and sensitive LAMP assay for MSV field detection. Ketsela et al. [20] confirmed the morphological symptoms of MSV infection chlorotic leaf streaks, chlorophyll loss, and growth retardation causing reduced yield or plant death through field observations. Finally, Wang et al. [21] demonstrated that stochastic models incorporating environmental transmission can explain periodic epidemic patterns, a concept relevant to multi-year epidemics of MSV. Recent studies have applied stochastic and delay-based epidemic modeling methods to a variety of infectious diseases, demonstrating the usefulness of dynamical consistency and global stability analyses for the interpretation of disease transmission and control measures [22,23]. Similar stochastic and bifurcation-based modeling methods have recently been formulated for human infectious diseases, such as influenza transmission and control, demonstrating the importance of stochastic effects and treatment–vaccination dynamics in epidemic models [24–27−]. Polynomial numerical schemes have proven successful in complex fractional dynamic systems, demonstrating Morgan Voyce polynomial approaches to time-fractional models [28,29]. Stochastic and cost-effectiveness modeling frameworks have been applied in recent epidemiological studies to examine intervention strategies for major infectious diseases such as HIV/AIDS and COVID-19, highlighting the applicability of data-driven approaches to the optimization of control strategies and vaccine efficacy [30–32].

Previous studies on Maize Streak Virus (MSV) dynamics have primarily relied on deterministic or fractional-order differential equation models, which, though useful, have a tendency to leave out the stochastic fluctuations and time-delays that occur in real-world agro-ecosystems. The majority of such studies have been idealized and have not tried to incorporate uncertainty due to environmental fluctuations, random infection, or heterogeneity in insect vector behavior. These limitations restrict their application to realistic field-scale epidemic prediction. To account for these lacunae, the present study develops a stochastic delayed model of MSV transmission incorporating randomness and temporal memory effects in plant-virus interactions. The study also provides a stochastic Nonstandard Finite Difference (NSFD) scheme, which is defined by positivity, boundedness, and step-size independence properties significant in the context of biological realism. The model combines theoretical analysis with computational efficiency and provides a more solid foundation for controlling MSV as well as other plant diseases.

The organization of the paper is as follows: Section 1 gives a review and thorough review of infectious maize streak disease-like disease reported in the literature. Sections 2 and 3 consider the establishment of the delayed model and the mathematical analysis later, and the two types of model equilibria and reproduction numbers. Sections 4 and 5 consider an investigation of the stochastic model, for example, its extinction and persistence. The stochastic NSFD approach is discussed in Section 6. Sections 7 and 8 are devoted particularly to numerical simulations and the presentation of results. Long-term opinions give a complete outline of the work under Section 9.

## 2 Formulation of model

This section presents a mathematical model describing the transmission dynamics of *Maize Streak Virus* (MSV) within a maize plant population. The total plant population at time $t$ is denoted by

$$N(t) = S(t) + F(t) + E(t) + I(t) + R(t),$$

where each compartment represents a distinct epidemiological state:

- $S(t)$—*Susceptible plants*: uninfected plants that can acquire infection after contact with infected plants or vectors.

- $F(t)$—*Insecticide-treated plants*: plants protected partially by insecticide application; protection depends on the insecticide's effectiveness.

- $E(t)$—*Exposed plants*: plants that have been infected but are in the latent (non-infectious) stage.

- $I(t)$—*Infected plants*: plants currently infectious and capable of transmitting MSV.

- $R(t)$—*Recovered plants*: plants that have gained temporary immunity following infection or treatment.

## 2.2 Model description

The dynamics of the system are governed by the following biological processes (Fig 1):

- Recruitment: Susceptible plants enter the population from the environment at a constant rate $\Lambda$.

- Infection: Healthy plants in the susceptible class $S(t)$ become exposed $E(t)$ after effective contact with infected plants $I(t)$. The disease transmission follows the law of mass action at a rate $\beta S(t)I(t)$.

- Insecticide application: A fraction of susceptible plants $S(t)$ is treated with insecticide at a rate $\theta$, moving them into the insecticide-treated class $F(t)$. The insecticide confers partial protection, determined by its efficacy $\gamma$.

- Loss of protection: Treated plants $F(t)$ gradually lose protection due to insecticide degradation or resistance and return to the susceptible class $S(t)$ at a rate $\varepsilon$.

- Exposure and infection: Exposed plants $E(t)$ progress to the infectious class $I(t)$ at a rate $\alpha$.

- Recovery: Infected plants $I(t)$ recover naturally or through treatment at a rate $\delta$, moving into the recovered class $R(t)$.

- Loss of immunity: Recovered plants $R(t)$ lose their immunity and return to the susceptible class $S(t)$ at a rate $\varphi$.

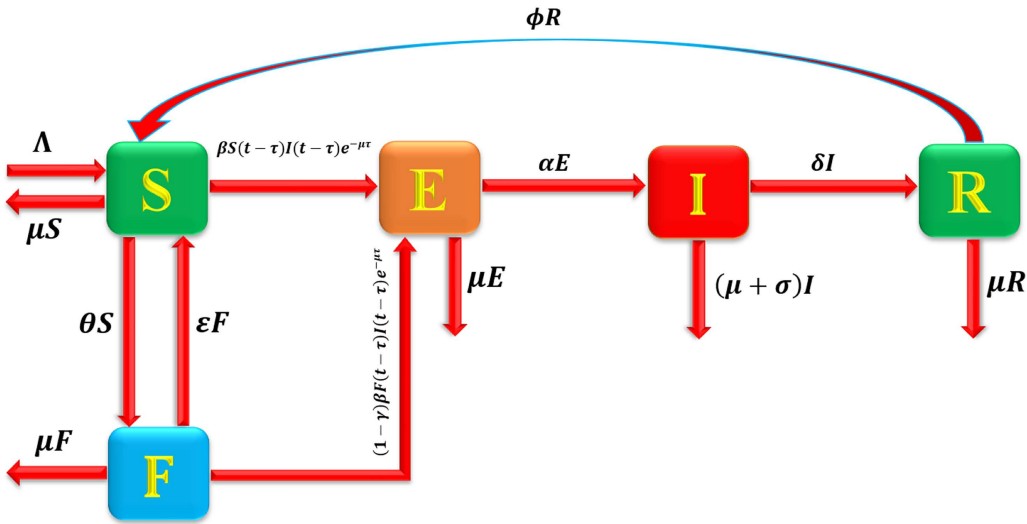

**Fig 1. Flow map of maize streak disease.**

- Secondary exposure: A proportion of insecticide-treated plants $F(t)$ may still become exposed $E(t)$ due to reduced insecticide efficacy or reinfection pressure.

The corresponding system of differential equations (not shown here) is derived based on these transition processes using the standard law of mass action.

$$\frac{dS(t)}{dt} = \Lambda - \beta S(t-\tau)I(t-\tau)e^{-\mu\tau} + \varepsilon F(t) + \phi R(t) - (\theta + \mu)S(t),$$
(1)

$$\frac{dF(t)}{dt} = \theta S(t) - (1-\gamma)\beta F(t-\tau)I(t-\tau)e^{-\mu\tau} - (\varepsilon + \mu)F(t),$$
(2)

$$\frac{dE(t)}{dt} = \beta S(t-\tau)I(t-\tau)e^{-\mu\tau} + (1-\gamma)\beta F(t-\tau)I(t-\tau)e^{-\mu\tau} - (\alpha + \mu)E(t),$$
(3)

$$\frac{dI(t)}{dt} = \alpha E(t) - (\delta + \sigma + \mu)I(t),$$
(4)

$$\frac{dR(t)}{dt} = \delta I(t) - (\phi + \mu)R(t),$$
(5)

where: $S(0) \geq 0$, $F(0) \geq 0$, $E(0) \geq 0$, $I(0) \geq 0$, $R(0) \geq 0$, and $t \geq 0$, $\tau < t$ are initial conditions.

## 3 Analysis of feasible properties

This section provides an analysis of the feasible properties of the stochastic delayed model (1)-(5).

### 3.1 Positivity and boundedness

To preserve the significant analysis of the model, each of the variables $S(t)$, $F(t)$, $E(t)$, $I(t)$, and $R(t)$ must be non-negative. That is, the outcomes of the model analysis at every time $t \geq 0$, $\tau < t$ in a practical range.

$$\mathcal{M} = \left\{ (S, F, E, I, R) \in \mathbb{R}_+^5 : N(t) \leq \frac{\Lambda}{\mu},\ S \geq 0, F \geq 0, E \geq 0, I \geq 0, R \geq 0 \right\}.$$

**Theorem 1.** (Positivity of Solutions): For the system (1)-(5), the solutions

$$(S(t), F(t), E(t), I(t), R(t)) \in \mathbb{R}_+^5$$

remain positive for all $t \geq 0$ and for all $\tau < t$, given non-negative initial conditions.

**Proof.** To show that each state variable remains non-negative, consider the system (1)-(5) and evaluate its right-hand sides on the corresponding boundary surfaces where one compartment equals zero while the others remain non-negative.

$$\begin{aligned}
\frac{dS}{dt}\Big|_{S=0} &= \Lambda + \varepsilon F + \varphi R > 0,\\
\frac{dF}{dt}\Big|_{F=0} &= \theta S(t) > 0,\\
\frac{dE}{dt}\Big|_{E=0} &= \beta S I e^{-\mu\tau} + (1-\gamma)\beta F I e^{-\mu\tau} > 0,\\
\frac{dI}{dt}\Big|_{I=0} &= \alpha E(t) > 0,\\
\frac{dR}{dt}\Big|_{R=0} &= \delta I(t) > 0.
\end{aligned}$$

Since all model parameters are positive and the inflow terms are non-negative, the vector field on each boundary of the positive orthant points inward. Consequently, solutions starting in $\mathbb{R}_+^5$ cannot cross the coordinate planes, ensuring that

$$(S(t), F(t), E(t), I(t), R(t)) \in \mathbb{R}_+^5 \,\forall\, t \geq 0.$$

Hence, the system (1)-(5) admits positive solutions for all $t \geq 0$, as required.

**Theorem 2.** (Boundedness of Solutions): For the system (1)-(5), the solutions

$$(S(t), F(t), E(t), I(t), R(t)) \in \mathbb{R}_+^5$$

are uniformly bounded for all $t \geq 0$.

Proof. Define the total population of maize plants as

$$N(t) = S(t) + F(t) + E(t) + I(t) + R(t).$$

By summing the differential equations (1)-(5), we obtain

$$\frac{dN}{dt} = \Lambda - \mu(S + F + E + I + R) = \Lambda - \mu N.$$

This differential inequality implies

$$\frac{dN}{dt} \leq \Lambda - \mu N.$$

Solving this inequality using the Grönwall lemma yields

$$N(t) \leq N(0)e^{-\mu t} + \frac{\Lambda}{\mu}(1 - e^{-\mu t}), t \geq 0.$$

As $t \to \infty$,

$$\limsup_{t \to \infty} N(t) \leq \frac{\Lambda}{\mu}.$$

Therefore, $N(t)$ - and consequently all state variables $S(t), F(t), E(t), I(t), R(t)$ remain bounded for all $t \geq 0$.

## 3.2 Model equilibria and reproduction number

The MS disease delayed model's equilibria will be briefly discussed in this section, and the maize streak free equilibrium ($MSFE - M_0$), and the maize streak endemic equilibrium ($MSEE - M^*$) will all be covered.

As

$$M_0 = (S_0, F_0, E_0, I_0, R_0) = \left( \frac{\Lambda(\varepsilon + \mu)}{(\varepsilon + \mu)(\theta + \mu) - \varepsilon\theta}, \frac{\Lambda\theta}{(\varepsilon + \mu)(\theta + \mu) - \varepsilon\theta}, 0, 0, 0 \right),$$

$$M^* = (S^*, F^*, E^*, I^*, R^*),$$

$$S^* = \frac{\Lambda + \varepsilon F^* + \phi R^*}{(\theta + \beta I^* e^{-\mu\tau} + \mu)}, F^* = \frac{\theta S^*}{((1-\gamma)\beta I^* e^{-\mu\tau} + \varepsilon + \mu)}, E^* = \frac{\beta S^* I^* e^{-\mu\tau} + (1-\gamma)\beta F^* I^* e^{-\mu\tau}}{(\alpha + \mu)},$$

$$I^* = \frac{\alpha E^*}{(\delta + \sigma + \mu)},$$

$$R^* = \frac{\delta I^*}{(\phi + \mu)}.$$

### 3.3 Basic reproduction number $R_0$

We compute $R_0$ via the next-generation matrix (NGM) method. The infected state vector is

$$\mathbf{x}(t) = \begin{pmatrix} E(t) \\ I(t) \end{pmatrix}.$$

Near the disease-free equilibrium (DFE) $M_0 = (S_0, F_0, E_0 = 0, I_0 = 0, R_0 = 0)$, the infection subsystem (with a fixed infection delay $\tau$) linearizes to

$$\begin{aligned} \dot{E}(t) &= \beta\, e^{-\mu\tau}(S_0 + (1-\gamma)F_0)\, I(t-\tau) - (\alpha + \mu)\, E(t), \\ \dot{I}(t) &= \alpha\, E(t) - (\delta + \sigma + \mu)\, I(t). \end{aligned}$$

**DFE values $(S_0, F_0)$**

At the DFE ($E = I = R = 0$) the susceptible insecticide subsystem satisfies

$$\begin{aligned} 0 &= \Lambda - (\theta + \mu)S_0 + \varepsilon F_0, \\ 0 &= \theta S_0 - (\varepsilon + \mu)F_0, \end{aligned}$$

which gives

$$S_0 = \frac{\Lambda(\varepsilon + \mu)}{\mu(\theta + \varepsilon + \mu)}, F_0 = \frac{\theta\Lambda}{\mu(\theta + \varepsilon + \mu)}.$$

**Next-generation matrices**

Write $\dot{\mathbf{x}} = \mathcal{F}(\mathbf{x}) - \mathcal{V}(\mathbf{x})$ with new-infection terms $\mathcal{F}$ and transition terms $\mathcal{V}$. The Jacobians at the DFE are

$$F = \left[\frac{\partial \mathcal{F}_i}{\partial x_j}\right]_{M_0} = \begin{pmatrix} 0 & \beta\, e^{-\mu\tau}(S_0 + (1-\gamma)F_0) \\ 0 & 0 \end{pmatrix}, V = \left[\frac{\partial \mathcal{V}_i}{\partial x_j}\right]_{M_0} = \begin{pmatrix} \alpha + \mu & 0 \\ -\alpha & \delta + \sigma + \mu \end{pmatrix}.$$

The next-generation matrix is $K = FV^{-1}$. Since

$$V^{-1} = \begin{pmatrix} \frac{1}{\alpha + \mu} & 0 \\ \frac{\alpha}{(\alpha + \mu)(\delta + \sigma + \mu)} & \frac{1}{\delta + \sigma + \mu} \end{pmatrix},$$

we obtain

$$K = \begin{pmatrix} \frac{\beta\, e^{-\mu\tau}(S_0 + (1-\gamma)F_0)\, \alpha}{(\alpha + \mu)(\delta + \sigma + \mu)} & \frac{\beta\, e^{-\mu\tau}(S_0 + (1-\gamma)F_0)}{\delta + \sigma + \mu} \\ 0 & 0 \end{pmatrix}.$$

The spectral radius of $K$ (largest eigenvalue) is therefore

$$R_0 = \frac{\beta\, e^{-\mu\tau}(S_0 + (1-\gamma)F_0)\, \alpha}{(\alpha + \mu)(\delta + \sigma + \mu)}.$$

## 4 Stability analysis

In this part, we study the stability of the model both locally and globally at its equilibrium point, with the findings proven in the established results as follows:

$$J_M = \begin{bmatrix} J_{11} & J_{12} & J_{13} & J_{14} & J_{15} \\ J_{21} & J_{22} & J_{23} & J_{24} & J_{25} \\ J_{31} & J_{32} & J_{33} & J_{34} & J_{35} \\ J_{41} & J_{42} & J_{43} & J_{44} & J_{45} \\ J_{51} & J_{52} & J_{53} & J_{54} & J_{55} \end{bmatrix},$$

(6)

$J_{11} = -\beta I e^{-\mu\tau} - (\theta + \mu), J_{12} = \varepsilon, J_{13} = 0, J_{14} = -\beta S e^{-\mu\tau}, J_{15} = \phi \ J_{21} = \theta, J_{22} = -(1-\gamma)\beta I e^{-\mu\tau} - (\varepsilon + \mu), J_{23} = 0,$
$J_{24} = -(1-\gamma)\beta F e^{-\mu\tau}, J_{25} = 0, J_{31} = \beta I e^{-\mu\tau}, J_{32} = (1-\gamma)\beta I e^{-\mu\tau}, J_{33} = -(\alpha + \mu), J_{34} = \beta S e^{-\mu\tau} + (1-\gamma)\beta F e^{-\mu\tau},$
$J_{35} = 0, \ J_{41} = 0, J_{42} = 0, J_{43} = \alpha, J_{44} = -(\delta + \sigma + \mu), J_{51} = 0, J_{52} = 0, J_{53} = 0, J_{54} = \delta, J_{55} = -(\phi + \mu).$

**Theorem 3. (Local Stability of the Disease-Free Equilibrium)**

The maize streak–free equilibrium

$$M_0 = (S_0, F_0, E_0, I_0, R_0)$$

of the system (1)-(5) is **locally asymptotically stable (LAS)** if the basic reproduction number $R_0 < 1$.

**Proof.** The Jacobian matrix of the system (1)-(5) evaluated at the equilibrium point $M_0$ is given by

$$J_M \mid_{M_0} = \begin{bmatrix} -(\theta + \mu) & \varepsilon & 0 & -\beta S_0 e^{-\mu\tau} & \varphi \\ \theta & -(\varepsilon + \mu) & 0 & -(1-\gamma)\beta F_0 e^{-\mu\tau} & 0 \\ 0 & 0 & -(\alpha + \mu) & \beta S_0 e^{-\mu\tau} + (1-\gamma)\beta F_0 e^{-\mu\tau} & 0 \\ 0 & \alpha & 0 & -(\delta + \sigma + \mu) & 0 \\ 0 & 0 & 0 & \delta & -(\varphi + \mu) \end{bmatrix}.$$

The characteristic equation associated with $J_M \mid_{M_0}$ is

$$\det(J_M \mid_{M_0} - \lambda I) = 0,$$

which can be written as the fifth-degree polynomial

$$\lambda^5 + A_4 \lambda^4 + A_3 \lambda^3 + A_2 \lambda^2 + A_1 \lambda + A_0 = 0,$$

where the coefficients $A_i (i = 0, \dots, 4)$ depend on model parameters as follows:

$$
\begin{aligned}
A_4 &= [(\alpha + \mu) + (\delta + \sigma + \mu) + (\varepsilon + \mu) + (\varphi + \mu)][1 + (\theta + \mu)], \\
A_3 &= ([(\alpha + \mu)(\delta + \sigma + \mu) - \alpha(\beta S_0 e^{-\mu\tau} + (1-\gamma)\beta F_0 e^{-\mu\tau})][(\alpha + \mu) + (\delta + \sigma + \mu)] \\
&\quad + [(\varepsilon + \mu) + (\varphi + \mu)](1 + (\theta + \mu))) - \theta\varepsilon, \\
A_2 &= ([(\varepsilon + \mu) + (\varphi + \mu)][(\alpha + \mu)(\delta + \sigma + \mu) - \alpha(\beta S_0 e^{-\mu\tau} + (1-\gamma)\beta F_0 e^{-\mu\tau})] \\
&\quad + [(\varepsilon + \mu) + (\varphi + \mu)][(\alpha + \mu) + (\delta + \sigma + \mu)])(1 + (\theta + \mu)) \\
&\quad - \theta\varepsilon[(\alpha + \mu) + (\delta + \sigma + \mu)] - \theta\varepsilon(\varphi + \mu), \\
A_1 &= (\alpha + \mu)(\delta + \sigma + \mu)[(\alpha + \mu)(\delta + \sigma + \mu) - \alpha(\beta S_0 e^{-\mu\tau} + (1-\gamma)\beta F_0 e^{-\mu\tau})](1 + (\theta + \mu)) \\
&\quad - \theta\varepsilon(\varphi + \mu)[(\alpha + \mu) + (\delta + \sigma + \mu)], \\
A_0 &= \alpha(\beta S_0 e^{-\mu\tau} + (1-\gamma)\beta F_0 e^{-\mu\tau}) - (\alpha + \mu)(\delta + \sigma + \mu) \\
&\quad + \theta\varepsilon\alpha(\beta S_0 e^{-\mu\tau} + (1-\gamma)\beta F_0 e^{-\mu\tau})(\varphi + \mu).
\end{aligned}
$$

(19)

For biologically feasible parameter values, all coefficients $A_i > 0$. Applying the Routh–Hurwitz stability criterion for a fifth-degree polynomial, the necessary and sufficient conditions for all roots to have negative real parts are satisfied when $R_0 < 1$. Hence, all eigenvalues of $J_M \mid_{M_0}$ possess negative real parts, and the maize streak–free equilibrium $M_0$ is locally asymptotically stable whenever $R_0 < 1$.

**Theorem 4. (Local Stability of the Maize Streak Endemic Equilibrium)**

The maize streak endemic equilibrium

$$M^* = (S^*, F^*, E^*, I^*, R^*)$$

of system (1)-(5) is **locally asymptotically stable (LAS)** if the basic reproduction number $R_0 > 1$.

**Proof.** The Jacobian matrix of system (1)-(5) evaluated at the endemic equilibrium $M^*$ is

$$J_M \mid_{M^*} = \begin{bmatrix} -\beta I^* e^{-\mu\tau} - (\theta + \mu) & \varepsilon & 0 & -\beta S^* e^{-\mu\tau} & \phi \\ \theta & -(1-\gamma)\beta I^* e^{-\mu\tau} - (\varepsilon + \mu) & 0 & -(1-\gamma)\beta F^* e^{-\mu\tau} & 0 \\ \beta I^* e^{-\mu\tau} & (1-\gamma)\beta I^* e^{-\mu\tau} & -(\alpha + \mu) & \beta S^* e^{-\mu\tau} + (1-\gamma)\beta F^* e^{-\mu\tau} & 0 \\ 0 & \alpha & 0 & -(\delta + \sigma + \mu) & 0 \\ 0 & 0 & 0 & \delta & -(\phi + \mu) \end{bmatrix}.$$

The characteristic equation associated with $J_M \mid_{M^*}$ is

$$\det(J_M \mid_{M^*} - \lambda I) = 0,$$

which can be written as the fifth-degree polynomial

$$\lambda^5 + A_4 \lambda^4 + A_3 \lambda^3 + A_2 \lambda^2 + A_1 \lambda + A_0 = 0,$$

where

$$
\begin{aligned}
A_4 &= (a_3 + a_8) + (a_1 + a_9) + a_7, \\
A_3 &= (a_3 a_8 + \alpha a_4) + (a_3 + a_8)(a_1 + a_9) + a_1 a_9 + (a_3 + a_8) + a_7(a_1 + a_9) - \varepsilon\theta, \\
A_2 &= (a_1 + a_9)(a_3 a_8 + \alpha a_4) + a_1 a_9(a_3 + a_8) + a_7(a_3 a_8 + \alpha a_4) + a_7(a_3 + a_8)(a_1 + a_9) \\
&\quad + a_1 a_7 a_9 - \varepsilon\theta(a_7 + a_9) - \varepsilon\theta a_8 - \theta\alpha a_2, \\
A_1 &= a_1 a_9(a_3 a_8 + \alpha a_4) + (a_3 a_8 + \alpha a_4)(a_1 + a_9)a_7 + a_1 a_7 a_9(a_3 + a_8) \\
&\quad - \varepsilon\theta a_7 a_9 - \varepsilon\theta a_8(a_7 + a_9) - \theta\alpha\varphi\delta, \\
A_0 &= a_1 a_7 a_9(a_3 a_8 + \alpha a_4) - \varepsilon\theta a_7 a_8 a_9 - \alpha\theta a_2 a_7 a_9 - a_7 \theta\alpha\varphi\delta,
\end{aligned}
$$

with parameter substitutions

$$
\begin{aligned}
a_1 &= \beta I^* e^{-\mu\tau} + (\theta + \mu), \; a_2 = \beta S^* e^{-\mu\tau}, \; a_3 = (1-\gamma)\beta I^* e^{-\mu\tau} + (\varepsilon + \mu), \\
a_4 &= (1-\gamma)\beta F^* e^{-\mu\tau}, \; a_5 = \beta I^* e^{-\mu\tau}, \; a_6 = (1-\gamma)\beta I^* e^{-\mu\tau}, \\
a_7 &= (\alpha + \mu), \; a_8 = (\delta + \sigma + \mu), \; a_9 = (\varphi + \mu).
\end{aligned}
$$

Since all biological parameters are positive, the coefficients $A_i$ $(i = 0, \ldots, 4)$ are positive and satisfy

$$A_4 A_3 > A_2, \; (A_4 A_3 - A_2)A_1 > (A_4 A_1 - A_0)A_2.$$

By the Routh–Hurwitz stability criterion for a fifth-degree polynomial, all eigenvalues of $J_M \mid_{M^*}$ have negative real parts when $R_0 > 1$.

Therefore, the maize streak endemic equilibrium $M^*$ is locally asymptotically stable whenever $R_0 > 1$.

**Theorem 5.** (Global Stability of the Disease-Free Equilibrium)

The disease-free equilibrium

$$M_0 = (S_0, F_0, E_0, I_0, R_0)$$

of system (1)-(5) is globally asymptotically stable (GAS) whenever $R_0 < 1$.

**Proof.** Consider the continuously differentiable Lyapunov function $U : M \to \mathbb{R}$ defined by

$$U = (S - S_0 - S_0 \ln \tfrac{S}{S_0}) + (F - F_0 - F_0 \ln \tfrac{F}{F_0}) + E + I + R.$$

Its time derivative along trajectories of system (1)-(5) is

$$\frac{dU}{dt} = \frac{S - S_0}{S} \frac{dS}{dt} + \frac{F - F_0}{F} \frac{dF}{dt} + \frac{dE}{dt} + \frac{dI}{dt} + \frac{dR}{dt}.$$

Substituting the corresponding right-hand sides of system (1)-(5) gives

$$\frac{dU}{dt} = \frac{S - S_0}{S}[\Lambda - \beta S I e^{-\mu\tau} + \varepsilon F + \varphi R - (\theta + \mu)S] + \frac{F - F_0}{F}[\theta S - (1 - \gamma)\beta F I e^{-\mu\tau} - (\varepsilon + \mu)F]$$
$$+ [\beta S I e^{-\mu\tau} + (1 - \gamma)\beta F I e^{-\mu\tau} - (\alpha + \mu)E] + [\alpha E - (\delta + \sigma + \mu)I] + [\delta I - (\varphi + \mu)R].$$

After algebraic simplification, we obtain

$$\frac{dU}{dt} = -(\Lambda + \varepsilon F + \varphi R)\frac{(S - S_0)^2}{S_0} - (\theta S)\frac{(F - F_0)^2}{F F_0} - (\sigma + \mu)I[1 - \frac{\beta S e^{-\mu\tau} + (1 - \gamma)\beta F e^{-\mu\tau}}{\sigma + \mu}] - \mu E - (\varphi + \mu)R.$$

All parameters are positive, and when $R_0 < 1$ the bracketed term is positive, ensuring that

$$\frac{dU}{dt} \le 0.$$

Equality $\frac{dU}{dt} = 0$ holds only at

$$S = S_0, \ F = F_0, \ E = I = R = 0.$$

By LaSalle's Invariance Principle, the largest invariant set where $\dot{U} = 0$ corresponds precisely to the equilibrium $M_0$. Therefore, every trajectory of the system tends to $M_0$ as $t \to \infty$.

Hence, $M_0$ is globally asymptotically stable whenever $R_0 < 1$.

**Theorem 6. (Global Stability of the Maize Streak Endemic Equilibrium)**

The maize streak endemic equilibrium

$$M^* = (S^*, F^*, E^*, I^*, R^*)$$

of system (1)-(5) is **globally asymptotically stable (GAS)** whenever $R_0 > 1$.

**Proof.** Define the continuously differentiable Lyapunov function $V : M \to \mathbb{R}$ as

$$V = (S - S^* - S^* \ln \tfrac{S}{S^*}) + (F - F^* - F^* \ln \tfrac{F}{F^*})$$
$$+ (E - E^* - E^* \ln \tfrac{E}{E^*}) + (I - I^* - I^* \ln \tfrac{I}{I^*}) + (R - R^* - R^* \ln \tfrac{R}{R^*}).$$

Its time derivative along the solutions of system (1)-(5) is given by

$$\frac{dV}{dt} = \frac{S-S^*}{S}\frac{dS}{dt} + \frac{F-F^*}{F}\frac{dF}{dt} + \frac{E-E^*}{E}\frac{dE}{dt} + \frac{I-I^*}{I}\frac{dI}{dt} + \frac{R-R^*}{R}\frac{dR}{dt}.$$

Substituting the model equations into (29) yields

$$\begin{aligned}\frac{dV}{dt} = &\frac{S-S^*}{S}[\Lambda - \beta S I e^{-\mu\tau} + \varepsilon F + \varphi R - (\theta + \mu)S]\\ &+ \frac{F-F^*}{F}[\theta S - (1-\gamma)\beta F I e^{-\mu\tau} - (\varepsilon + \mu)F]\\ &+ \frac{E-E^*}{E}[\beta S I e^{-\mu\tau} + (1-\gamma)\beta F I e^{-\mu\tau} - (\alpha + \mu)E]\\ &+ \frac{I-I^*}{I}[\alpha E - (\delta + \sigma + \mu)I] + \frac{R-R^*}{R}[\delta I - (\varphi + \mu)R].\end{aligned}$$

After simplification, we obtain

$$\frac{dV}{dt} = -(\Lambda + \varepsilon F + \varphi R)\frac{(S-S^*)^2}{SS^*} - (\theta S)\frac{(F-F^*)^2}{FF^*} - (\beta S I e^{-\mu\tau} + (1-\gamma)\beta F I e^{-\mu\tau})\frac{(E-E^*)^2}{EE^*} - (\alpha E)\frac{(I-I^*)^2}{II^*} - (\delta I)\frac{(R-R^*)^2}{RR^*}.$$

All model parameters and equilibrium components are positive. When $R_0 > 1$, each term is non-positive, ensuring that $\frac{dV}{dt} \le 0$. Equality $\frac{dV}{dt} = 0$ holds only when

$$S = S^*, F = F^*, E = E^*, I = I^*, R = R^*.$$

By **LaSalle's Invariance Principle**, the only invariant set contained in $\{\dot{V} = 0\}$ corresponds to the equilibrium $M^*$. Therefore, all trajectories of the system approach $M^*$ as $t \to \infty$. Hence, the maize streak endemic equilibrium $M^*$ is **globally asymptotically stable** whenever $R_0 > 1$.

## 5 Stochastic formulation Phase 1

Based on the model (1)-(5), consider a vector $\mathcal{W} = [S(t), F(t), E(t), I(t), R(t)]^T$ of stochastic delay differential equations (SDDEs). Calculating the variance $\mathcal{E}^*\left[\Delta U \left(\Delta U\right)^T\right]$ and expectations $\mathcal{E}^*[\Delta U]$ is our goal. Table 1 lists the likelihood of changes together with the corresponding transition time.

$$\text{Expectation} = \mathcal{E}^*[\Delta U] = \sum_{i=1}^{13}\mathcal{P}_i(\Delta U)_i = \begin{bmatrix} \Lambda - \beta S I e^{-\mu\tau} + \varepsilon F + \phi R - (\theta + \mu)S \\ \theta S - (1-\gamma)\beta F I e^{-\mu\tau} - (\varepsilon + \mu)F \\ \beta S I e^{-\mu\tau} + (1-\gamma)\beta F I e^{-\mu\tau} - (\alpha + \mu)E \\ \alpha E - (\delta + \sigma + \mu)I \\ \delta I - (\phi + \mu)R \end{bmatrix} \Delta t.$$

$$\text{Variance} = \sum_{i=1}^{13}\mathcal{P}_i(\Delta\mathfrak{U})_i\left[(\Delta\mathfrak{U})_i\right]^T$$

$$= \begin{bmatrix} P_1 + P_2 + P_3 + P_4 + P_5 + P_6 & -P_3 - P_5 & -P_2 & 0 & -P_4 \\ -P_3 - P_5 & P_3 + P_5 + P_7 + P_8 & -P_7 & 0 & 0 \\ -P_2 & -P_7 & P_2 + P_7 + P_9 + P_{10} & -P_9 & 0 \\ 0 & 0 & -P_9 & P_9 + P_{11} + P_{12} & -P_{11} \\ -P_4 & 0 & 0 & -P_{11} & P_4 + P_{11} + P_{13} \end{bmatrix} \Delta t,$$

$$\text{Drift} = \mathcal{G}(\mathfrak{U}, t) = \frac{\mathcal{E}^*[\Delta U]}{\Delta t} = \begin{bmatrix} \Lambda - \beta S I e^{-\mu\tau} + \varepsilon F + \phi R - (\theta + \mu)S \\ \theta S - (1-\gamma)\beta F I e^{-\mu\tau} - (\varepsilon + \mu)F \\ \beta S I e^{-\mu\tau} + (1-\gamma)\beta F I e^{-\mu\tau} - (\alpha + \mu)E \\ \alpha E - (\delta + \sigma + \mu)I \\ \delta I - (\phi + \mu)R \end{bmatrix} \Delta t,$$

$$(7)$$

**Table 1. Illustrates an implicit modification to the model's process.**

| Transition | Probabilities |
|---|---|
| $(\Delta \mathfrak{U})_1 = \begin{bmatrix} 1 & 0 & 0 & 0 & 0 \end{bmatrix}^{\mathsf{T}}$ | $P_1 = (\Lambda)\Delta t$ |
| $(\Delta \mathfrak{U})_2 = \begin{bmatrix} -1 & 0 & 1 & 0 & 0 \end{bmatrix}^{\mathsf{T}}$ | $P_2 = (\beta S I e^{-\mu\tau})\,\Delta t$ |
| $(\Delta \mathfrak{U})_3 = \begin{bmatrix} 1 & -1 & 0 & 0 & 0 \end{bmatrix}^{\mathsf{T}}$ | $P_3 = (\varepsilon F)\Delta t$ |
| $(\Delta \mathfrak{U})_4 = \begin{bmatrix} 1 & 0 & 0 & 0 & -1 \end{bmatrix}^{\mathsf{T}}$ | $P_4 = (\phi R)\Delta t$ |
| $(\Delta \mathfrak{U})_5 = \begin{bmatrix} -1 & 1 & 0 & 0 & 0 \end{bmatrix}^{\mathsf{T}}$ | $P_5 = (\theta S)\Delta t$ |
| $(\Delta \mathfrak{U})_6 = \begin{bmatrix} -1 & 0 & 0 & 0 & 0 \end{bmatrix}^{\mathsf{T}}$ | $P_6 = (\mu S)\Delta t$ |
| $(\Delta \mathfrak{U})_7 = \begin{bmatrix} 0 & -1 & 1 & 0 & 0 \end{bmatrix}^{\mathsf{T}}$ | $P_7 = ((1-\gamma)\beta F I e^{-\mu\tau})\,\Delta t$ |
| $(\Delta \mathfrak{U})_8 = \begin{bmatrix} 0 & -1 & 0 & 0 & 0 \end{bmatrix}^{\mathsf{T}}$ | $P_8 = (\mu F)\Delta t$ |
| $(\Delta \mathfrak{U})_9 = \begin{bmatrix} 0 & 0 & -1 & 1 & 0 \end{bmatrix}^{\mathsf{T}}$ | $P_9 = (\alpha E)\Delta t$ |
| $(\Delta \mathfrak{U})_{10} = \begin{bmatrix} 0 & 0 & -1 & 0 & 0 \end{bmatrix}^{\mathsf{T}}$ | $P_{10} = (\mu E(t))\,\Delta t$ |
| $(\Delta \mathfrak{U})_{11} = \begin{bmatrix} 0 & 0 & 0 & -1 & 1 \end{bmatrix}^{\mathsf{T}}$ | $P_{11} = (\delta I)\Delta t$ |
| $(\Delta \mathfrak{U})_{12} = \begin{bmatrix} 0 & 0 & 0 & -1 & 0 \end{bmatrix}^{\mathsf{T}}$ | $P_{12} = ((\sigma + \mu)I)\,\Delta t$ |
| $(\Delta \mathfrak{U})_{13} = \begin{bmatrix} 0 & 0 & 0 & 0 & -1 \end{bmatrix}^{\mathsf{T}}$ | $P_{13} = (\mu R)\Delta t$ |

$$\text{Diffusion} = \mathcal{H}(\mathfrak{U}, t) = \sqrt{\frac{\mathcal{E}^*\left[\Delta\mathfrak{U}\,(\Delta\mathfrak{U})^{\mathsf{T}}\right]}{\Delta t}},$$

$$= \sqrt{\begin{bmatrix} P_1 + P_2 + P_3 + P_4 + P_5 + P_6 & -P_3 - P_5 & -P_2 & 0 & -P_4 \\ -P_3 - P_5 & P_3 + P_5 + P_7 + P_8 & -P_7 & 0 & 0 \\ -P_2 & -P_7 & P_2 + P_7 + P_9 + P_{10} & -P_9 & 0 \\ 0 & 0 & -P_9 & P_9 + P_{11} + P_{12} & -P_{11} \\ -P_4 & 0 & 0 & -P_{11} & P_4 + P_{11} + P_{13} \end{bmatrix}} \quad (8)$$

Therefore, $d\mathfrak{U}(t) = G(\mathfrak{U}, t) + H(\mathfrak{U}, t)\,dB(t)$.

$$d\begin{bmatrix} S \\ F \\ E \\ I \\ R \end{bmatrix} = \begin{bmatrix} \Lambda - \beta S I e^{-\mu\tau} + \varepsilon F + \phi R - (\theta + \mu)S \\ \theta S - (1-\gamma)\beta F I e^{-\mu\tau} - (\varepsilon + \mu)F \\ \beta S I e^{-\mu\tau} + (1-\gamma)\beta F I e^{-\mu\tau} - (\alpha + \mu)E \\ \alpha E - (\delta + \sigma + \mu)I \\ \delta I - (\phi + \mu)R \end{bmatrix} dt + $$
$$\sqrt{\begin{bmatrix} P_1 + P_2 + P_3 + P_4 + P_5 + P_6 & -P_3 - P_5 & -P_2 & 0 & -P_4 \\ -P_3 - P_5 & P_3 + P_5 + P_7 + P_8 & -P_7 & 0 & 0 \\ -P_2 & -P_7 & P_2 + P_7 + P_9 + P_{10} & -P_9 & 0 \\ 0 & 0 & -P_9 & P_9 + P_{11} + P_{12} & -P_{11} \\ -P_4 & 0 & 0 & -P_{11} & P_4 + P_{11} + P_{13} \end{bmatrix}}\,dB(t). \quad (9)$$

The next section will cover the conventional numerical methods for approximating solutions to Stochastic models. We all concur that $I_n = \{0, 1, 2, 3, ..., n\}$. If $N \in \mathbb{N}$, then the temporal interval [0, T] is consistently divided with a uniform partition equal to $\tau = \frac{T}{N}$, and the corresponding nodes are given as $0 = t_0 < t_1 < t_2 < ... < t_N = T$.

For each $n \in I_N$. Further, this will be agreed by us $\mathfrak{U}_n = \mathfrak{U}_{t_N}$, however, $n \in I_N$ and $\mathfrak{U}(t) = (S, F, E, I, R)^t$, $\Delta W_n = W(t_n + 1) - W(t_n)$.

The academic literature on the subject is consulted to simulate its results of Eq. (9) using the Euler-Maruyama approach. The details are displayed in Table 1 and are as follows;

$$\mathfrak{U}_{n+1} = \mathfrak{U}_n + \mathcal{G}(\mathfrak{U}_n, t)\, \Delta t + \mathcal{H}(\mathfrak{U}_n, t)\, dB(t).$$

$$\begin{bmatrix} S^{n+1} \\ F^{n+1} \\ E^{n+1} \\ I^{n+1} \\ R^{n+1} \end{bmatrix} = \begin{bmatrix} S^n \\ F^n \\ E^n \\ I^n \\ R^n \end{bmatrix} + \begin{bmatrix} \Lambda - \beta S I e^{-\mu\tau} + \varepsilon F + \phi R - (\theta + \mu)S \\ \theta S - (1-\gamma)\beta F I e^{-\mu\tau} - (\varepsilon + \mu)F \\ \beta S I e^{-\mu\tau} + (1-\gamma)\beta F I e^{-\mu\tau} - (\alpha + \mu)E \\ \alpha E - (\delta + \sigma + \mu)I \\ \delta I - (\phi + \mu)R \end{bmatrix} \Delta t$$

$$+ \sqrt{\begin{bmatrix} P_1 + P_2 + P_3 + P_4 + P_5 + P_6 & -P_3 - P_5 & -P_2 & 0 & -P_4 \\ -P_3 - P_5 & P_3 + P_5 + P_7 + P_8 & -P_7 & 0 & 0 \\ -P_2 & -P_7 & P_2 + P_7 + P_9 + P_{10} & -P_9 & 0 \\ 0 & 0 & -P_9 & P_9 + P_{11} + P_{12} & -P_{11} \\ -P_4 & 0 & 0 & -P_{11} & P_4 + P_{11} + P_{13} \end{bmatrix}} \Delta B_n,$$

(10)

where the discretization parameter is indicated by $\Delta t$.

## 6 Stochastic formulation Phase 2

By incorporating Brownian motion, we get the dynamical system unreliable parameters (1)-(5). In the sequence described below:

$$\frac{dS(t)}{dt} = \Lambda - \beta S(t)I(t)e^{-\mu\tau} + \varepsilon F(t) + \phi R(t) - (\theta + \mu)S(t) + \sigma_1 S(t)dB(t),$$

(11)

$$\frac{dF(t)}{dt} = \theta S(t) - (1-\gamma)\beta F(t)I(t)e^{-\mu\tau} - (\varepsilon + \mu)F(t) + \sigma_2 F(t)dB(t),$$

(12)

$$\frac{dE(t)}{dt} = \beta S(t)I(t)e^{-\mu\tau} + (1-\gamma)\beta F(t)I(t)e^{-\mu\tau} - (\alpha + \mu)E(t) + \sigma_3 E(t)dB(t),$$

(13)

$$\frac{dI(t)}{dt} = \alpha E(t) - (\delta + \sigma + \mu)I(t) + \sigma_4 IdB(t),$$

(14)

$$\frac{dR(t)}{dt} = \delta I(t) - (\phi + \mu)R(t) + \sigma_5 RdB(t),$$

(15)

where the uncertainty of each compartment and existence of $B(t)$ Brownian motion are denoted by $\sigma_i; i = 1, 2, 3, 4, 5$.

### 6.1 Feasible properties

This model (11)–(15) concludes the examination of the positivity and boundedness features of the system.

Let us assume the following vector:

$$\mathcal{V}(t) = (S(t), F(t), E(t), I(t), R(t)),$$

and norm

$$|\mathcal{V}(t)| = \sqrt{S^2(t) + F^2(t) + E^2(t) + I^2(t) + R^2(t)}. \tag{16}$$

Moreover, let $\mathcal{D}_1^{4,1}\left(\mathbb{R}^5 x(0,\infty) : \mathbb{R}_+\right)$ represent the set of all positive functions $\mathcal{U}_1\left(\mathcal{V},t\right)$ that are subsequently defined on $\mathbb{R}^5 x(0,\infty)$. Furthermore, in $\mathcal{V}$, the function is once differentiable and twice differentiable. We have defined differentiable. We have defined the differentiable operator $\mathcal{T}_1$, which is associated with stochastic delay differential equations (SDDEs) in four dimensions.

$$d\mathcal{V}(t) = \mathcal{D}_1\left(\mathcal{V},t\right)dt + \mathcal{K}_1\left(\mathcal{V},t\right)dB(t). \tag{17}$$

As,

$$\mathcal{T}_1 = \frac{\partial}{\partial t} + \sum_{i=1}^{5}\mathcal{D}_{1i}\left(\mathcal{V},t\right)\frac{\partial}{\partial \mathcal{V}_i} + \frac{1}{2}\sum_{i,j=1}^{5}\mathcal{K}_1^{\ T}\left(\mathcal{V},t\right)\mathcal{K}_1\left(\mathcal{V},t\right)\frac{\partial^2}{\partial \mathcal{U}_i\partial \mathcal{U}_j}.$$

If $\mathcal{T}_1$ acts on function $\mathcal{V}^* \in \mathcal{D}_1^{4,1}\left(\mathbb{R}^4 x(0,\infty) : \mathbb{R}_+\right)$ then we denote

$$\mathcal{T}_1\mathcal{V}^*\left(\mathcal{V},t\right) = \mathcal{V}_t^*\left(\mathcal{V},t\right) + \mathcal{V}_\mathcal{V}^*\left(\mathcal{V},t\right)\mathcal{D}_1\left(\mathcal{V},t\right) + \frac{1}{2}\mathit{Trace}\left(\mathcal{K}_1^{\ T}\left(\mathcal{V},t\right)\mathcal{V}_{\mathcal{V}\mathcal{V}}^*\left(\mathcal{V},t\right)\mathcal{K}_1\left(\mathcal{U},t\right)\right),$$

where $\mathcal{T}$ is Transportation.

**Theorem 7:** Demonstrates that there exists only one solution $(S(t),F(t),E(t),I(t),R(t))$ for the system (11)–(15) for all initial conditions $(S(0),F(0),E(0),I(0),R(0)) \in \mathbb{R}_+^5$. With a probability of one, these solutions will also invariably stay in $\mathbb{R}_+^5$.

*Proof.* Given that all model parameters are locally satisfiable by the Lipschitz bounds. Therefore, based on Ito's formula, the above model has a positive solution locally on the interval $[0,\ \ell_e]$, and $\ell_e$ is the timing of the explosion. The global solution of the model can be shown when $\ell_e$ equals infinity.

Define $\mathfrak{g}_0 = 0$ to be a big enough number so that $S(0),F(0),E(0),I(0)$, and $R(0)$ are all included in the interval $\left\{\frac{1}{\mathfrak{g}_0},\mathfrak{g}_0\right\}$. Let's construct the subsequent sequence for each positive integer "$\mathfrak{g}$".

$$\ell_n = \inf\left\{t \in [0,\ \ell_e] : S(t) \in \left(\frac{1}{\mathfrak{g}},\mathfrak{g}\right), F(t) \in \left(\frac{1}{\mathfrak{g}},\mathfrak{g}\right), or\ E(t) \in \left(\frac{1}{\mathfrak{g}},\mathfrak{g}\right), or\ I(t) \in \left(\frac{1}{\mathfrak{g}},\mathfrak{g}\right),\ or\ R(t) \in \left(\frac{1}{\mathfrak{g}},\mathfrak{g}\right)\right\}, \tag{18}$$

where we set $\inf\varphi = \infty(\varphi$ *is the empty set*$)$. Since $\ell_n$ is non-decreasing as $n \to \infty$,

$$\ell_\infty = \lim_{n\to\infty}\ell_n. \tag{19}$$

According to the inequality, $\ell_\infty$ is either equal to or smaller than $\ell_e$. Our goal now is to show that, as we intended, $\ell_\infty$ equals infinity.

When $\mathcal{T} > 0$ and $\mathscr{b}_1 \in (0,\ 1)$ are found, the statement is satisfied. If this condition is not met.

$$\mathfrak{U}\left\{\ell_n \leq \mathcal{T}\right\} \geq \mathscr{b}_1 \qquad \forall b \geq \mathscr{b}_1. \tag{20}$$

Define a $\mathcal{C}^4-$function $f : \mathbb{R}_+^4 \to \mathbb{R}_+$ by

$$\mathscr{f}(S,F,E,I,R) = (S-1-\ln S) + (F-1-\ln F) + (E-1-\ln E) + (I-1-\ln I) + (R-1-\ln R). \tag{21}$$

By using Ito's formula, we calculate

$$d\mathscr{f}(S,F,E,I,R) = \left(1-\frac{1}{S}\right)dS + \left(1-\frac{1}{F}\right)dF + \left(1-\frac{1}{E}\right)dE + \left(1-\frac{1}{I}\right)dI + \left(1-\frac{1}{R}\right)dR + \frac{\sigma_1^2+\sigma_2^2+\sigma_3^2+\sigma_4^2+\sigma_5^2}{2}dt.$$

$$df(S, F, E, I, R) = \left(1 - \tfrac{1}{S}\right)\left((\Lambda - \beta S(t)I(t)e^{-\mu\tau} + \varepsilon F(t) + \phi R(t) - (\theta + \mu)S(t)\right)dt$$
$$+\sigma_1 S(t)dB(t) + \left(1 - \tfrac{1}{F}\right)\left((\theta S(t) - (1-\gamma)\beta F(t)I(t)e^{-\mu\tau} - (\varepsilon + \mu)F(t)\right)dt$$
$$+\sigma_2 F(t)dB(t) + \left(1 - \tfrac{1}{E}\right)\left((\beta S(t)I(t)e^{-\mu\tau} + (1-\gamma)\beta F(t)I(t)e^{-\mu\tau} - (\alpha + \mu)E(t)\right)dt$$
$$+\sigma_3 E(t)dB(t) + \left(1 - \tfrac{1}{I}\right)\left((\alpha E(t) - (\delta + \sigma + \mu)I(t))dt + \sigma_4 I(t)dB(t)\right) + \left(1 - \tfrac{1}{R}\right)\left((\delta I(t) - (\phi + \mu)R(t)\right)dt$$
$$+\sigma_5 R(t)dB(t) + \tfrac{\sigma_1^2 + \sigma_2^2 + \sigma_3^2 + \sigma_4^2 + \sigma_5^2}{2}dt.$$

$$df(S, F, E, I, R) = \left(\Lambda + \theta + 5\mu + \varepsilon + \alpha + \delta + \sigma + \phi + \tfrac{\sigma_1^2 + \sigma_2^2 + \sigma_3^2 + \sigma_4^2 + \sigma_5^2}{2}\right)dt$$
$$+ \sigma_1 S(t)dB(t) + \sigma_2 F(t)dB(t) + \sigma_3 E(t)dB(t) + \sigma_4 I(t)dB(t) + \sigma_5 R(t)dB(t). \tag{22}$$

To simplify, we assume $\mathcal{P}_1 = \left(\Lambda + \theta + 5\mu + \varepsilon + \alpha + \delta + \sigma + \phi + \tfrac{\sigma_1^2 + \sigma_2^2 + \sigma_3^2 + \sigma_4^2 + \sigma_5^2}{2}\right)$, then Eq. (22) could be written as:

$$df(S, E, I, R) \leq \mathcal{P}_1 dt + [\sigma_1 S(t) + \sigma_2 F(t) + \sigma_3 E(t) + \sigma_4 I(t) + \sigma_5 R(t)]\, d(B(t)), \tag{23}$$

where $\mathcal{P}_1$ is a positive constant, after integrating from 0 to $\ell_n \wedge I$, we get,

$$\int_0^{\ell_n \wedge I} df(S, F, E, I, R) \leq \int_0^{\ell_n \wedge I} \mathcal{P}_1 ds + \int_0^{\ell_n \wedge I} [\sigma_1 S(t) + \sigma_2 F(t) + \sigma_3 E(t) + \sigma_4 I(t) + \sigma_5 R(t)]\, d(B(t)), \tag{24}$$

where $\ell_n \wedge I = \min(\ell_n, \mathcal{T})$, taking the expectations leads to

$$\mathcal{E}\mathcal{V}^*\left(S(\ell_n \wedge I), F(\ell_n \wedge I), E(\ell_n \wedge I), I(\ell_n \wedge I), R(\ell_n \wedge I)\right) \leq \mathcal{V}^*\left(S(0), F(0), E(0), I(0), R(0)\right) + \mathcal{P}_1 \mathcal{T}. \tag{25}$$

Set $\mathfrak{S}_n = \{\ell_n \leq \mathcal{T}\}$ for $n > n_1$ and from (19), we have $\mathcal{X}(\mathfrak{S}_n \geq b)$.

For each element $a_1$ in the set $\mathfrak{S}_n$, there exist certain indices $i$ such that $\mathcal{V}_i(\ell_n, a_1)$ is equal to either $n$ or $\tfrac{1}{n}$, where $i$ takes on the values 1, 2, 3, 4, and 5.

Hence, $\mathcal{V}^*\left((S(\ell_n, a_1), F(\ell_n, a_1), E(\ell_n, a_1), I(\ell_n, a_1), R(\ell_n, a_1))\right)$ is less than $\min\left\{n - 1 - lnn, \tfrac{1}{n} - 1 - ln\tfrac{1}{n}\right\}$.
Next, we obtain

$$\mathcal{V}^*\left(S(0), F(0), E(0), I(0), R(0)\right) + \mathcal{P}_1 \mathcal{T} \geq \mathcal{E}\left(I_{\mathfrak{S}_n(a_1)}\mathcal{V}^*\left(S(\ell_n), F(\ell_n), E(\ell_n), I(\ell_n), R(\ell_n)\right)\right) \geq$$
$$\min\left\{n - 1 - lnn, \tfrac{1}{n} - 1 - ln\tfrac{1}{n}\right\}. \tag{26}$$

The indicator function is denoted as $I_{\mathfrak{S}_n(a_1)}$ within the set $\mathfrak{S}_n$. As $n$ approaches infinity, we get to the contradiction that infinity is equal to the value of $\mathcal{V}^*\left(S(0), F(0), E(0), I(0), R(0)\right) + \mathcal{P}_1 \mathcal{T}$, which is finite, as desired.

**Theorem 8.** If the spectral radius and the variance $\sigma_4^2 < \frac{\beta S_0 e^{-\mu\tau} + (1-\gamma)\beta F_0 e^{-\mu\tau}}{(\alpha + \mu)(\delta + \sigma + \mu)}$, then the number of infected plants in the system (11)-(15) will exponentially approach zero.

**Proof:** Let's examine the initial data $(S(0), F(0), E(0), I(0), R(0)) \in \mathbb{R}_+^5$ and the system (11)-(15) has a solution $(S(t), F(t), E(t), I(t), R(t))$ if it satisfies the stochastic delayed differential equation, where $\sigma_4$ represents randomness and $c$ represents drift.

$$dI(t) = \left(\frac{\beta S(t)I(t)e^{-\mu\tau} + (1-\gamma)\beta F(t)I(t)e^{-\mu\tau}}{(\alpha + \mu)} - (\delta + \sigma + \mu)I(t)\right)dt + c\sigma_4 I(t)dB(t),$$

Applying Ito's lemma to the function $g(I) = ln(I)$, we obtain

$$dln(I(t)) = g'(I(t))\, dI + \frac{1}{2}g''(I)I^2\sigma_4^2 dt,$$

$$dln(I(t)) = \frac{1}{I(t)}dI + \frac{1}{2}\left(-\frac{1}{I^2}\right)I^2\sigma_4^2 dt,$$

$$dln(I(t)) = \frac{1}{I(t)}dI - \frac{1}{2}\sigma_4^2 dt,$$

$$dln(I(t)) = \frac{1}{I(t)}\left[\left(\frac{\beta S(t)I(t)e^{-\mu\tau} + (1-\gamma)\beta F(t)I(t)e^{-\mu\tau}}{(\alpha+\mu)} - (\delta+\sigma+\mu)I(t)\right)dt + c\sigma_4 I(t)dB(t)\right] - \frac{1}{2}\sigma_4^2 dt,$$

$$dln(I(t)) = \left(\frac{\beta S(t)e^{-\mu\tau} + (1-\gamma)\beta F(t)e^{-\mu\tau}}{(\alpha+\mu)} - (\delta+\sigma+\mu)\right)dt + c\sigma_4 dB(t) - \frac{1}{2}\sigma_4^2 dt,$$

$$ln(I(t)) = lnI(0) + \left(\frac{\beta S(t)e^{-\mu\tau} + (1-\gamma)\beta F(t)e^{-\mu\tau}}{(\alpha+\mu)} - (\delta+\sigma+\mu) - \frac{1}{2}\sigma_4^2\right)dt + \int_0^t c\sigma_4 dB(t).$$

Notice that $N(t) = \int_0^t c\sigma_4 dB(t)$ with $N(0) = 0$.

If $\sigma_4^2 > \frac{\beta S_0 e^{-\mu\tau} + (1-\gamma)\beta F_0 e^{-\mu\tau}}{(\alpha+\mu)(\delta+\sigma+\mu)}$,

$$ln(I(t)) > \left(\frac{\beta S(t)e^{-\mu\tau} + (1-\gamma)\beta F(t)e^{-\mu\tau}}{(\alpha+\mu)} - (\delta+\sigma+\mu) - \frac{1}{2}\frac{\beta S_0 e^{-\mu\tau} + (1-\gamma)\beta F_0 e^{-\mu\tau}}{(\alpha+\mu)(\delta+\sigma+\mu)}\right)t + N(t) + lnI(0),$$

$$\frac{lnI(t)}{t} > \left(\frac{1}{2}\frac{\beta S_0 e^{-\mu\tau} + (1-\gamma)\beta F_0 e^{-\mu\tau}}{(\alpha+\mu)(\delta+\sigma+\mu)} - (\delta+\sigma+\mu)\right) + \frac{N(t)}{t} + \frac{lnI(0)}{t},$$

$$\lim_{t\to\infty}\frac{lnI(t)}{t} > \left(\frac{1}{2}\frac{\beta S_0 e^{-\mu\tau} + (1-\gamma)\beta F_0 e^{-\mu\tau}}{(\alpha+\mu)(\delta+\sigma+\mu)} - (\delta+\sigma+\mu)\right) > 0, \text{ with } \lim_{t\to\infty}\frac{N(t)}{t} = 0,$$

If $\sigma_2^2 < \frac{\beta S_0 e^{-\mu\tau} + (1-\gamma)\beta F_0 e^{-\mu\tau}}{(\alpha+\mu)(\delta+\sigma+\mu)}$, then

$$ln(I(t)) < \left(\frac{\beta S_0 e^{-\mu\tau} + (1-\gamma)\beta F_0 e^{-\mu\tau}}{(\alpha+\mu)(\delta+\sigma+\mu)} - (\delta+\sigma+\mu) - \frac{1}{2}\sigma_4^2\right)t + N(t) + lnI(0),$$

$$\frac{lnI(t)}{t} < (\delta+\sigma+\mu)\left(\frac{\beta S_0 e^{-\mu\tau} + (1-\gamma)\beta F_0 e^{-\mu\tau}}{(\alpha+\mu)(\delta+\sigma+\mu)} - 1 - \frac{1}{2}\sigma_4^2\right) + \frac{N(t)}{t} + \frac{lnI(0)}{t},$$

$\lim\sup_{t\to\infty}\frac{lnI(t)}{t} < (\delta+\sigma+\mu)\left(R_0^S - 1\right)$, when $R_0^S < 1$, we get $\lim\sup_{t\to\infty}\frac{lnI(t)}{t} \leq 0$,

$\lim_{t\to\infty} I(t) = 0$, as desired,

$$R_0^S = R_0^d - \frac{\sigma_4^2}{2(\delta+\sigma+\mu)} < 1.$$

## 7 Numerical methodology

Let $\mathcal{U}_n$ be the set defined for each $e \in \mathbb{N}$ as $\mathcal{U}_e = \{0, 1, 2, ..., e\}$. In this section, we will denote and analyze a discretization of the system (11)-(15). To achieve the objective, we consider the temporal interval where $T > 0$. Create a consistent partition of the time interval [0, T] into n subintervals, with a length of $k = \frac{T}{e}$ for each subinterval. For each $a \in I_e$, where $I_e$ is the collection of indices by considering $t_a = ak$. The numerical approximations for the functions S, F, E, I, and R are denoted as $S^n, F^n, E^n, I^n$, and $R^n$, respectively. The discrete initial data $(S_0, F_0, E_0, I_0, R_0)$ is defined, satisfying $S_0 = S(0), F_0 = F(0), E_0 = E(0), I_0 = I(0), R_0 = R(0)$ as required.

### 7.1 Stochastic nonstandard computational method

In our first equation (11) of the parametric perturbation model can be expressed with a non-standard computing approach; namely, equations (11)-(15) might be solved with a stochastic non-standard finite difference method.

$$dS(t) = \left(\Lambda - \beta SIe^{-\mu\tau} + \varepsilon F + \phi R - (\theta + \mu)S\right) dt + \sigma_1 Sd\left(B(t)\right). \tag{27}$$

For the stochastic NSFD approach

$$\frac{S^{n+1} - S^n}{h} = \left[\Lambda - \beta S^{n+1}I^n e^{-\mu\tau} + \varepsilon F^n + \phi R^n - (\theta + \mu)S^{n+1} + \sigma_1 S^n \Delta B_n\right]. \tag{28}$$

The system (11)-(15) can be decomposed by the stochastic NSFD process, as indicated in (28), and the entire system can then be expressed as follows:

$$S^{n+1} = \frac{S^n + h\left[\Lambda + \varepsilon F^n + \phi R^n + \sigma_1 S^n \Delta B_n\right]}{1 + h\left(\beta I^n e^{-\mu\tau} + (\theta + \mu)\right)}, \tag{29}$$

$$F^{n+1} = \frac{F^n + h\left[\theta S^n + \sigma_2 F^n \Delta B_n\right]}{1 + h\left((1-\gamma)\beta I^n e^{-\mu\tau} + (\varepsilon + \mu)\right)}, \tag{30}$$

$$E^{n+1} = \frac{E^n + h\left[\beta S^n I^n e^{-\mu\tau} + (1-\gamma)\beta F^n I^n e^{-\mu\tau} + \sigma_3 E^n \Delta B_n\right]}{1 + h(\alpha + \mu)}, \tag{31}$$

$$I^{n+1} = \frac{I^n + h\left[\alpha E^n + \sigma_4 I^n \Delta B_n\right]}{1 + h(\delta + \sigma + \mu)}, \tag{32}$$

$$R^{n+1} = \frac{R^n + h\left[\delta I^n + \sigma_5 R^n \Delta B_n\right]}{1 + h(\phi + \mu)}, \tag{33}$$

where, $n = 0, 1, 2, \ldots$ and $\Delta B_n = \Delta B_{t_{n+1}} - \Delta B_{t_n}$ is a general normal distribution, i.e., $\Delta B_n \sim N(0, 1)$.

## 7.2 Convergence analysis

The following theorems are stated concerning the convergence analysis.

**Theorem 8:** There is only one positive solution $(S, F, E, I, R) \in \mathbb{R}^5_+, \forall n > 0$ for any initial value $(S(0), F(0), E(0), I(0), R(0)) \in \mathbb{R}^5_+$ for equations (29) through (33).

***Proof:*** The evidence is verifiable since the non-positive property of the biological problems' constraint facilitates ease in demonstration.

**Theorem 9:** For the region
$\mathcal{M} = \left\{(S^n, F^n, E^n, I^n, R^n) \in \mathbb{R}^5_+ : S^n + F^n + E^n + I^n + R^n = N \leq \frac{\Lambda}{\mu}, \ S^n \geq 0, F^n \geq 0, E^n \geq 0, I^n \geq 0, R^n \geq 0\right\}$. For every $n \geq 0$ is an area of equations that is feasible and positively invariant (29) to (33).

***Proof:*** The system (29) to (33) can be deconstructed and considered $\Delta B_n = 0$, as follows:

$$\frac{S^{n+1} - S^n}{h} = \Lambda - \beta S^{n+1}I^n e^{-\mu\tau} + \varepsilon F^n + \phi R^n - (\theta + \mu)S^{n+1},$$

$$\frac{F^{n+1} - F^n}{h} = \theta S^n - (1-\gamma)\beta F^{n+1}I^n e^{-\mu\tau} - (\varepsilon + \mu)F^{n+1},$$

$$\frac{E^{n+1} - E^n}{h} = \beta S^n I^n e^{-\mu\tau} + (1-\gamma)\beta F^n I^n e^{-\mu\tau} - (\alpha + \mu)E^{n+1},$$

$$\frac{I^{n+1} - I^n}{h} = \alpha E^n - (\delta + \sigma + \mu)I^{n+1},$$

$$\frac{R^{n+1} - R^n}{h} = \delta I^n - (\phi + \mu)R^{n+1}.$$

Next, we get

$$\frac{(S^{n+1} + F^{n+1} + E^{n+1} + I^{n+1} + R^{n+1}) - (S^n + F^n + E^n + I^n + R^n)}{h} \leq \Lambda - \mu(S^n + F^n + E^n + I^n + R^n),$$

$$(S^{n+1} + F^{n+1} + E^{n+1} + I^{n+1} + R^{n+1}) - (S^n + F^n + E^n + I^n + R^n) \leq h\Lambda - h\mu(S^n + F^n + E^n + I^n + R^n),$$

$$(S^{n+1} + F^{n+1} + E^{n+1} + I^{n+1} + R^{n+1}) - (S^n + F^n + E^n + I^n + R^n) \leq h\Lambda - h\mu(S^n + F^n + E^n + I^n + R^n),$$

$(S^{n+1} + F^{n+1} + E^{n+1} + I^{n+1} + R^{n+1}) \leq \frac{\Lambda}{\mu}$, as desired.

**Theorem 10:** The suggested computational method is stable for any $n > 0$ if the eigenvalue is located in the unit circle.

***Proof:*** Let the function $Y$, $G$, $H$, $P$, and $Q$, which are the right-hand sides of the equations (29–33). Consider $\Delta B_n = 0$.

Here,

$Y = \frac{S + h[\Lambda + \varepsilon F + \phi R]}{1 + h(\beta I e^{-\mu\tau} + (\theta + \mu))}$, $G = \frac{F + h[\theta S]}{1 + h((1-\gamma)\beta I e^{-\mu\tau} + (\varepsilon + \mu))}$, $H = \frac{E + h[\beta S I e^{-\mu\tau} + (1-\gamma)\beta F I e^{-\mu\tau}]}{1 + h(\alpha + \mu)}$,

$P = \frac{I + h[\alpha E]}{1 + h(\delta + \sigma + \mu)}$, $Q = \frac{R + h[\delta I]}{1 + h(\phi + \mu)}$.

It is well known that a system of the forms (29–33) converges to the optimal state of the model if and only if the spectral radius of the Jacobian, (J),

$$J = \begin{bmatrix} \frac{\partial Y}{\partial S} & \frac{\partial Y}{\partial F} & \frac{\partial Y}{\partial E} & \frac{\partial Y}{\partial I} & \frac{\partial Y}{\partial R} \\ \frac{\partial G}{\partial S} & \frac{\partial G}{\partial F} & \frac{\partial G}{\partial E} & \frac{\partial G}{\partial I} & \frac{\partial G}{\partial R} \\ \frac{\partial H}{\partial S} & \frac{\partial H}{\partial F} & \frac{\partial H}{\partial E} & \frac{\partial H}{\partial I} & \frac{\partial H}{\partial R} \\ \frac{\partial P}{\partial S} & \frac{\partial P}{\partial F} & \frac{\partial P}{\partial E} & \frac{\partial P}{\partial I} & \frac{\partial P}{\partial R} \\ \frac{\partial Q}{\partial S} & \frac{\partial Q}{\partial F} & \frac{\partial Q}{\partial E} & \frac{\partial Q}{\partial I} & \frac{\partial Q}{\partial R} \end{bmatrix}. \tag{34}$$

For the stability of the model. It follows the conditions:

When $\rho(J) < 1$, the model's equilibrium is stable. The stability of the model's equilibria depends on whether $\rho(J) > 1$. The model's equilibria are naturally stable when $\rho(J) = 1$.

The components of the method-related Jacobian can be expressed as follows: maize streak-free equilibrium, $M_0 = (S_0, F_0, E_0, I_0, R_0)$.

$$J(M_0) = \begin{bmatrix} \frac{1}{1+h(\theta+\mu)} & 0 & 0 & -\frac{(S_0 + h[\Lambda + \varepsilon F_0 + \phi R])(h\beta e^{-\mu\tau})}{(1+h(\theta+\mu))^2} & \frac{h[\phi]}{1+h(\theta+\mu)} \\ \frac{h[\theta S_0]}{1+h(\varepsilon+\mu)} & \frac{1-h[\varepsilon](h[\theta S_0])}{1+h(\varepsilon+\mu)} & 0 & -\frac{(F_0 + h[\theta S_0])((1-\gamma)h\beta e^{-\mu\tau})}{(1+h(\varepsilon+\mu))^2} & 0 \\ 0 & 0 & \frac{1}{1+h(\alpha+\mu)} & \frac{h[\beta S_0 e^{-\mu\tau} + (1-\gamma)\beta F_0 e^{-\mu\tau}]}{1+h(\alpha+\mu)} & 0 \\ 0 & 0 & \frac{h[\alpha]}{1+h(\delta+\sigma+\mu)} & \frac{1}{1+h(\delta+\sigma+\mu)} & 0 \\ 0 & 0 & 0 & \frac{h[\delta]}{1+h(\phi+\mu)} & \frac{1}{1+h(\phi+\mu)} \end{bmatrix}.$$

So, the eigenvalues of the Jacobian at $M_0$ as follows:

$$\lambda_1 = \frac{1 - h[\varepsilon]\,(h\,[\theta S_0])}{1 + h(\varepsilon + \mu)} < 1, \lambda_2 = \frac{1}{1 + h(\theta + \mu)} < 1, \lambda_3 = \frac{1}{1 + h(\phi + \mu)}.$$

$$\begin{vmatrix} \frac{1}{1+h(\alpha+\mu)} & \frac{h[\beta S_0 e^{-\mu\tau} + (1-\gamma)\beta F_0 e^{-\mu\tau}]}{1+h(\alpha+\mu)} \\ \frac{h[\alpha]}{1+h(\delta+\sigma+\mu)} & \frac{h[\delta]}{1+h(\phi+\mu)} \end{vmatrix} = 0.$$

$$A_1 = \text{Trce of J}(M_0) = \frac{1}{1 + h(\alpha + \mu)} + \frac{h[\delta]}{1 + h(\phi + \mu)},$$

$$A_2 = \text{Determinent of } J(M_0) = \left( \left( \frac{1}{1+h(\alpha+\mu)} \right) \left( \frac{h[\delta]}{1+h(\phi+\mu)} \right) \right) + \left( \left( \frac{h\left[\beta S_0 e^{-\mu\tau} + (1-\gamma)\beta F_0 e^{-\mu\tau}\right]}{1+h(\alpha+\mu)} \right) \left( \frac{h[\alpha]}{1+h(\delta+\sigma+\mu)} \right) \right).$$

*Lemma.* For the quadratic equation $\lambda_2 - A_1\lambda + A_2 = 0$, $|\lambda_i| < 1$, $i = 1, 2$ if and only if the following conditions are satisfied:

i. $1 + A_1 + A_2 > 0$.

ii. $1 - A_1 + A_2 > 0$.

iii. $A_2 < 1$.

*Proof.* The proof is straightforward.

## 8 Computational results

In this section, we compare conventional numerical methods with a non-conventional computational technique to evaluate their efficiency, precision, and computational cost in solving the proposed model. This consideration highlights the advantages and potential drawbacks of using nonstandard numerical methods in advanced epidemiological models. Parameter estimates used in this work were drawn in great part directly from the literature, making proper consistency with earlier validated MSV transmission models. A few parameters (e.g., recruitment rate $\Lambda$, natural mortality rate $\mu$, and noise intensity $\sigma$) were assumed to be in biologically reasonable ranges since there is no special experimental data. All parameters are displayed in Table 2. All simulations were conducted using identical parameter sets (Table 2) and a fixed time horizon $t=500$. The deterministic solution was used as the benchmark reference. The stochastic NSFD method demonstrates the best overall performance, achieving lower error and faster computation while maintaining numerical stability independent of step size (see Table 3).

### 8.1 Discussion

This section covered the discussion of graphical representations of the behavior of infected plants for different time step sizes $(h)$ with time delay $(\tau = 1)$. Fig 2 shows the infected plants over time at the endemic equilibrium point with a step size of $h = 0.01$. The model uses the Euler-Maruyama method, the stochastic NSFD method, and a deterministic approach. The small step size allows for more precise modeling of infection dynamics, capturing subtle changes in infection rates. In Fig 2, the stochastic nature leads to small fluctuations around the deterministic trajectory, which stabilizes as time progresses. Maize Streak Disease has a regular pattern with minimal variability, suggesting that for instances

**Table 2. Parameter values used in the model.**

| Parameter | Description | Value | Source/ Assumption |
|---|---|---|---|
| $\Lambda$ | Recruitment rate of susceptible plants | 0.5 | Assumed within biological range |
| $\beta$ | Transmission rate | 0.018 | Literature [1] |
| $\theta$ | Insecticide application rate | 0.1 | Literature [1] |
| $\mu$ | Natural mortality rate of plants | 0.5 | Assumed |
| $\varepsilon$ | Rate of loss of insecticide protection | 0.001 | Literature [1] |
| $\delta$ | Recovery rate of infected plants | 0.03 | Literature [1] |
| $\gamma$ | Efficacy of insecticide treatment | 0.9998 | Literature [1] |
| $\alpha$ | Progression rate from exposed to infectious class | 0.1 | Literature [1] |
| $\phi$ | Rate of loss of immunity | 0.015 | Literature [1] |
| $\sigma$ | Noise intensity in stochastic model | 0.02 | Assumed |

**Table 3. Quantitative comparison of stochastic numerical methods.**

| Numerical Method | CPU Time (s) | Mean Squared Error (MSE) | Maximum Stable Step Size ($h_{max}$) | Remarks |
|---|---|---|---|---|
| Stochastic Euler–Maruyama | 4.82 | $2.41 \times 10^{-3}$ | 0.5 | Stable only for small $h$; exhibits oscillations for large delays |
| Stochastic Runge–Kutta | 5.37 | $1.86 \times 10^{-3}$ | 0.6 | Moderate accuracy; partial loss of positivity for $h > 0.6$ |
| Stochastic NSFD (proposed) | 3.15 | $0.97 \times 10^{-3}$ | Step-size independent | Preserves positivity and stability for all tested $h$ values |

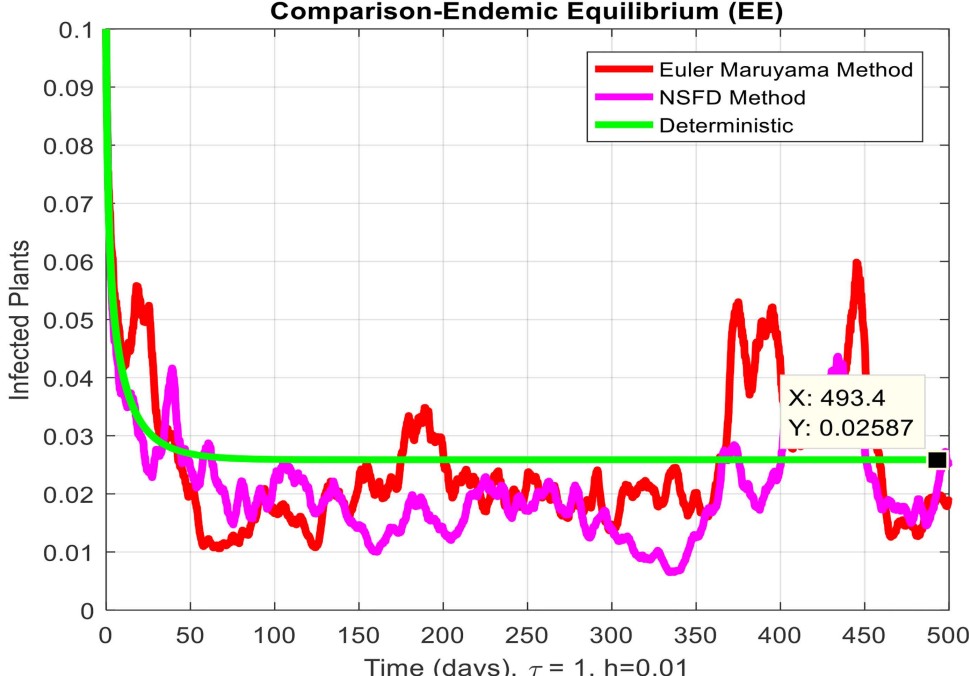

**Fig 2. Infected plants at EE when** $h = 0.01$.

with minimal environmental or vector behavior variation, the infection is quite under control. By boosting the step size to h = 1, Fig 3 creates larger stochastic fluctuations. With bigger steps, the infection graph appears to be more unpredictable, particularly during the initial stage of the epidemic. The stochastic Euler-Maruyama and stochastic NSFD methods are the ones that have more variability than the deterministic model. This conveys that with less time resolution (larger delays or less frequent observations), the disease spread is less predictable with more chance of larger outbreaks. The step size increment shows the stochastic effects amplification, that is, environmental noise (for instance, abrupt temperature changes or vector activity) can cause fluctuations in the infection rate that are more than the normal by a large margin. Likewise, Fig 4 presents the same with a step size of h = 0.01 but it emphasizes the Euler method of stochastic modeling. The rate of infection is rather stable at endemic equilibrium and very little variability occurs. The Maize Streak Disease secures a uniform pattern, and the stochastic methods' variability is undetectable, which means that the disease is kept under control in very strict and well-monitored conditions. In Fig 5 the step size is set to h = 1 and the variability of infection is significantly increased. The infected plant populations are showing larger fluctuations, with the stochastic methods (stochastic Euler and stochastic NSFD) straying quite a bit from the deterministic trajectory. This infers that

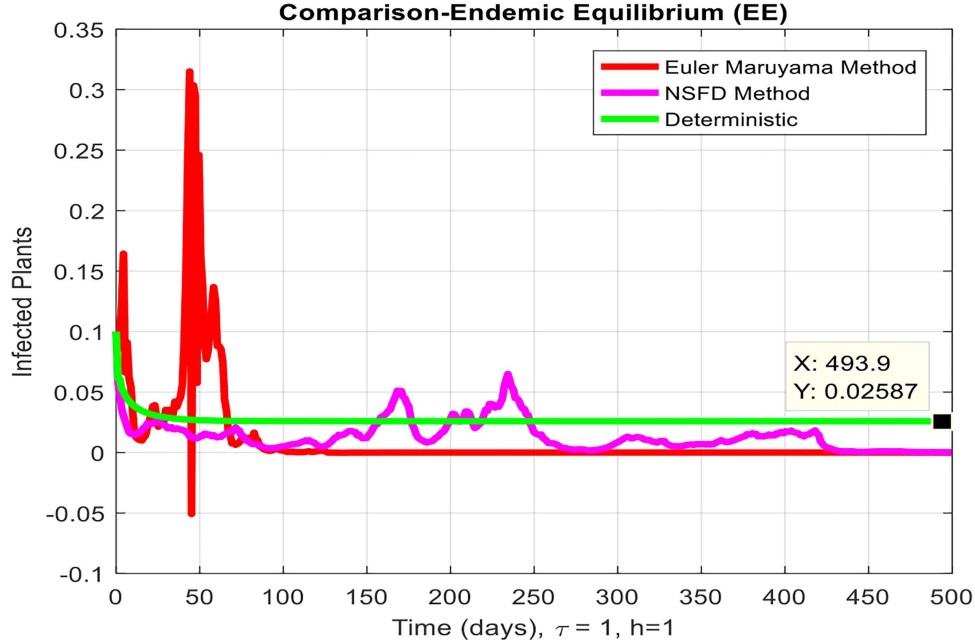

**Fig 3. Infected plants at EE when $h = 1$.**

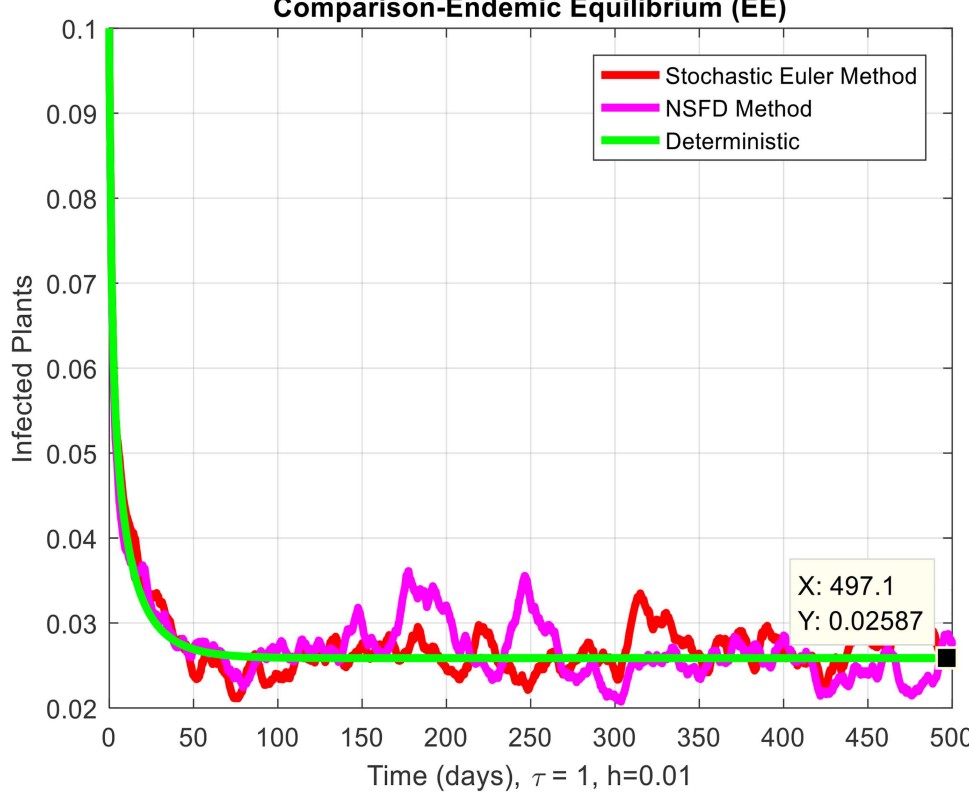

**Fig 4. Infected plants at EE when $h = 0.01$.**

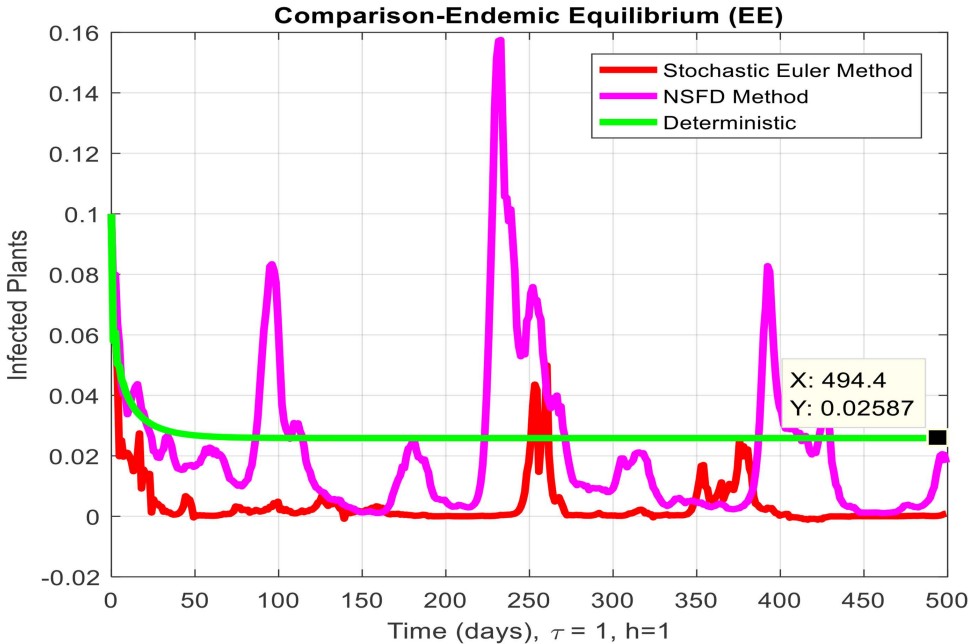

**Fig 5. Infected plants at EE when $h = 1$.**

in situations where there are delayed responses or more coarse time scales, the disease can fluctuate more severely, thereby making the control more challenging. The increase in step size illustrates how delay or inaccurate monitoring can result in ever more random outcomes. The stochastic Runge-Kutta method is being used in Fig 6 with a small step size

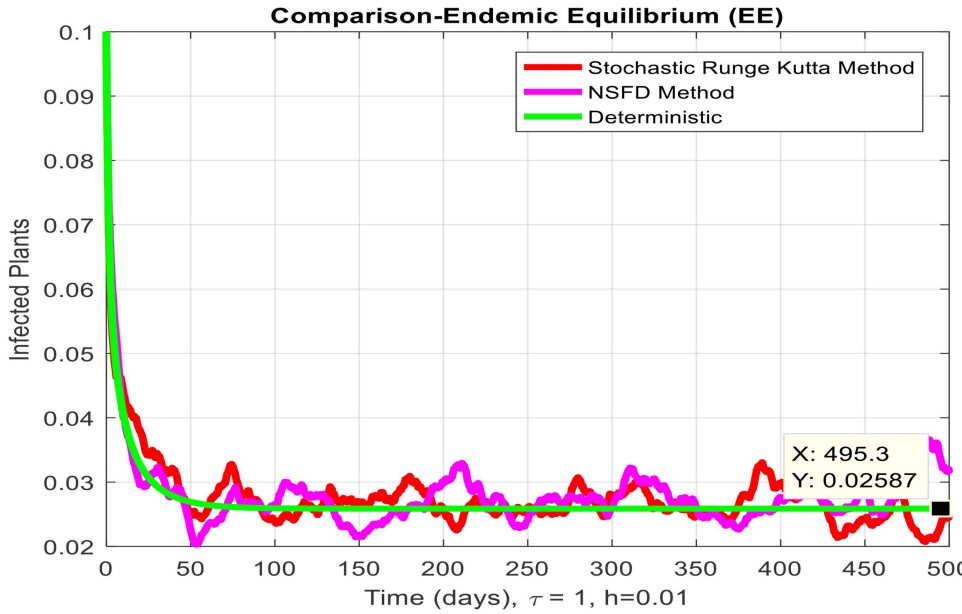

**Fig 6. Infected plants at EE when $h = 0.01$.**

(h = 0.01). The infection curve is unchanged with slight variations, just like in the other figures with smaller step sizes. The dissemination of Maize Streak Disease is under control, and there is no considerable stochastic impact. The stochastic Runge-Kutta method, like the Euler-Maruyama method, shows very low variability at this step size, which gives further evidence that the disease can be controlled fairly well if closely monitored. In Fig 7, where the step size is larger, h = 1, more variability is seen in the infection rate. The infection curve becomes less smooth, showing that the progress of the disease is more difficult to predict for larger steps. Delays in observation or in intervention might lead to greater outbreaks. Fig 8 illustrates how different delays affect the number of susceptible plants over time. Four different delays are shown: $\tau = 0.1, 0.3, 0.5,$ and $0.7$. As the delay increases, the number of susceptible plants drops more slowly, indicating that larger delays in implementing control measures allow more plants to remain vulnerable to infection. This explains that effective interventions happen to be those at appropriate times for the prevention of the disease by Maize Streak Disease. In Fig 9 infected plant population in a function of time with different time delays. Different values of time delay between $\tau = 0.1$ and $\tau = 0.7$. As the delay increases, the rate of acceleration of infected plants increases and keeps increasing with time. This would be interpreted as the longer duration held in the interventions of the disease reveals more severe outbreaks of Maize Streak Disease. Shorter delays translated to quicker recovery of the disease ($\tau = 0.1$), however, the longer delays showed constant outbreaks and higher infection rates. The illustration above stresses how timely interventions are essential to control the transmission of the disease. Fig 10 shows how the delay ($\tau$) affects the basic reproduction number ($R_0$). We can observe that as ($\tau$) increases, so does $R_0$. An increase in $R_0$ certainly tells one that the disease becomes more transmissible as the delay increases. After introducing the delays into the control measures of Maize Streak Disease, therefore, it can infect more plants and make the disease generally more transmissible. Without delays ($\tau = 0$), $R_0$ is relatively low, which means that this disease is more easily controlled. Increasing ($\tau$) causes $R_0$ to rise thus, the disease starts to be uncontrollable.

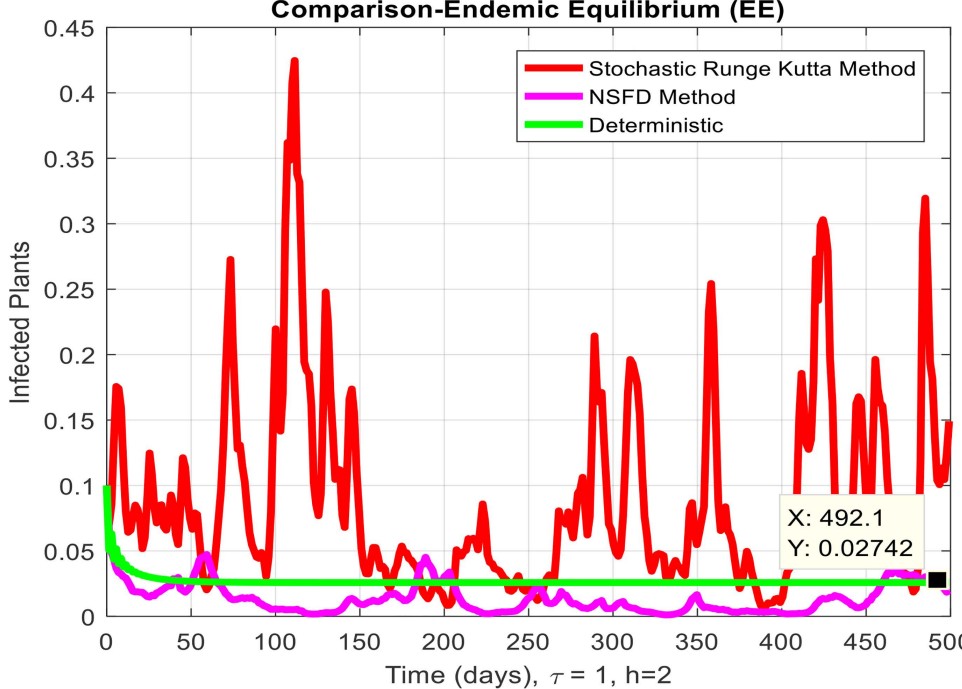

**Fig 7. Infected plants at EE when $h = 2$.**

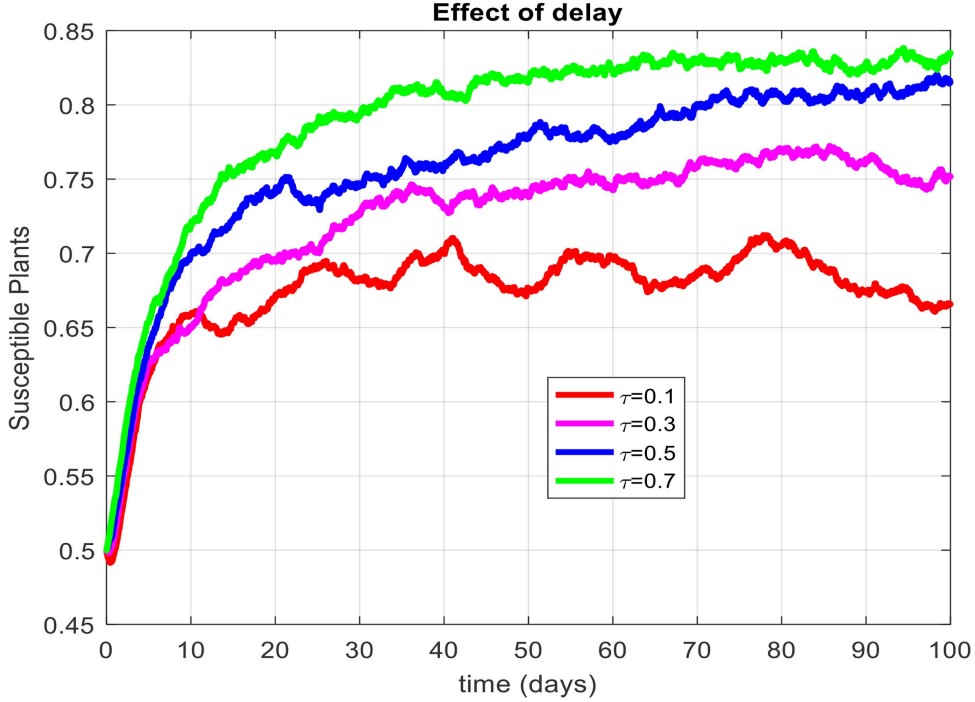

**Fig 8. Effect of delay on susceptible plants.**

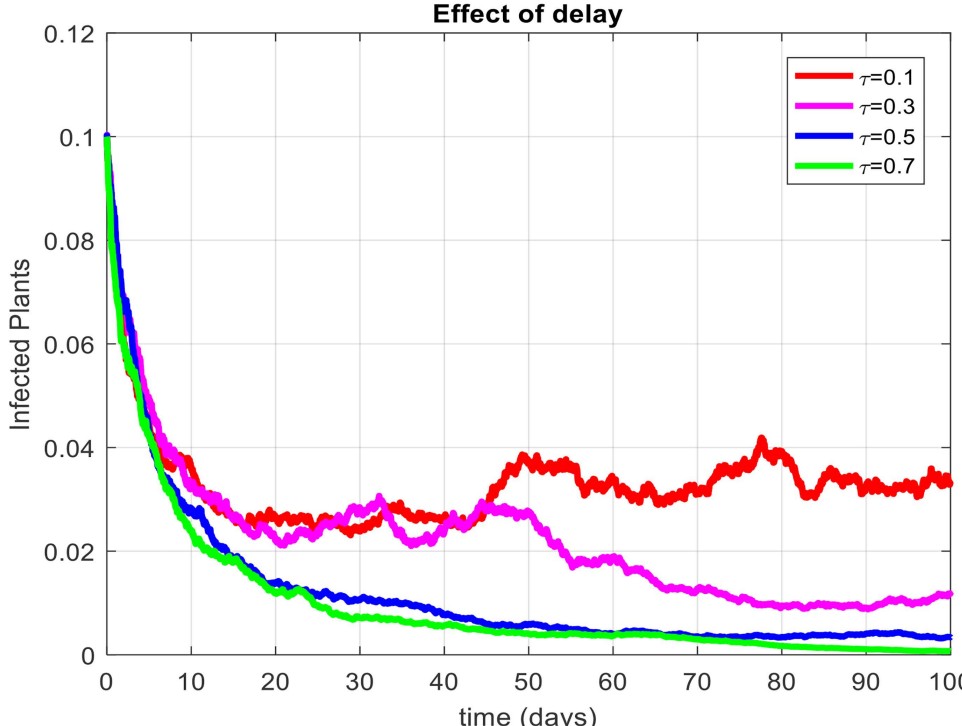

**Fig 9. Effect of delay on infected plants.**

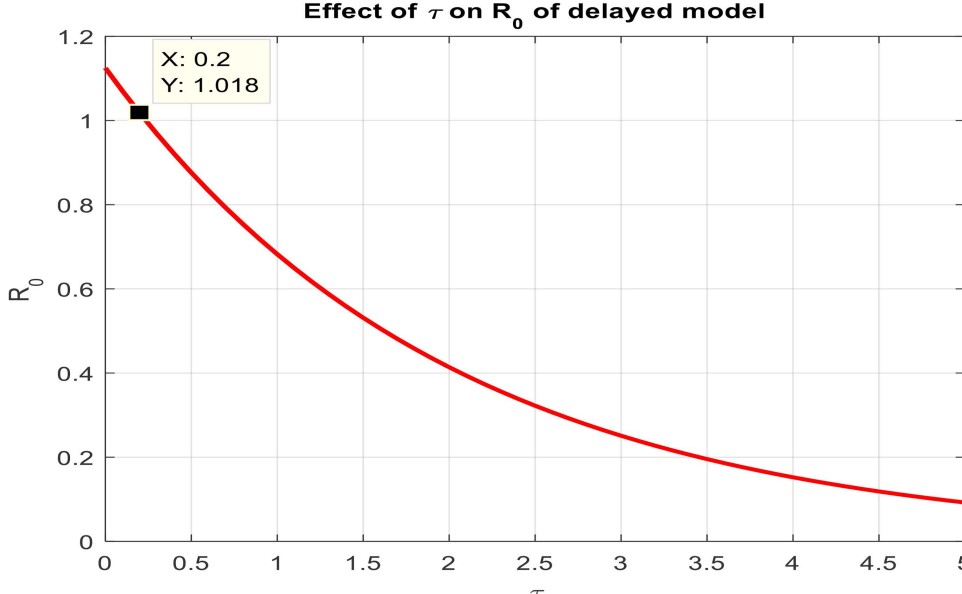

**Fig 10. Effect of $\tau$ on $R_0$ of the delayed model.**

## 9 Conclusion

This paper formulates a computational and mathematical study of the MSV disease transmission model of time-delay dynamics. Maize plants were divided into five compartments: susceptible, insecticide-treated, recovered, infected, and exposed. Dynamical analysis considered key epidemiological features such as the basic reproduction number, equilibria, boundedness, and positivity conditions. Both the local and global stability of the endemic and maize streak–free equilibrium points were investigated, establishing the asymptotic behavior of the system under various parameter conditions. The stochastic generalizations of the model, such as positivity, extinction, and persistence of the disease under stochastic perturbation, were also studied. Among the computational methods attempted, the stochastic Nonstandard Finite Difference (NSFD) scheme offered improved accuracy, stability, and biological consistency performance. In particular, stability is crucial in stochastic epidemic modeling to prevent spurious oscillations and numerical instability. Compared to other conventional numerical methods such as the stochastic Runge–Kutta, stochastic Euler, and Euler–Maruyama schemes, the stochastic NSFD method-maintained robustness and consistency even at large simulation times. Overall, the results confirm that the stochastic NSFD method provides a reliable, efficient, and biologically sound model for describing complex plant viral epidemics like MSV, with implications for future research and disease management practices.

### Author contributions

**Conceptualization:** Ali Raza, Nauman Ahmed.

**Data curation:** Ali Raza, Marek Lampart, Nauman Ahmed.

**Formal analysis:** Naveed Shahid, Ali Raza.

**Funding acquisition:** Ali Raza, Hala H. Taha.

**Investigation:** Sana Iqbal, Naveed Shahid, Ali Raza.

**Methodology:** Ali Raza, Marek Lampart, Dumitru Baleanu.

**Project administration:** Ali Raza, Marek Lampart, Nauman Ahmed.

**Resources:** Ali Raza, Nauman Ahmed.

**Software:** Ali Raza, Nauman Ahmed.

**Supervision:** Ali Raza, Marek Lampart, Nauman Ahmed, Dumitru Baleanu.

**Validation:** Ali Raza, Dumitru Baleanu.

**Visualization:** Ali Raza, Hala H. Taha.

**Writing – original draft:** Sana Iqbal, Ali Raza.

**Writing – review & editing:** Ali Raza, Nauman Ahmed.

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
