## [Decision Letter · Decision Letter 0]

26 Oct 2025

Dear Dr. Raza,

Thank you for submitting your manuscript to PLOS ONE. After careful consideration, we feel that it has merit but does not fully meet PLOS ONE’s publication criteria as it currently stands. Therefore, we invite you to submit a revised version of the manuscript that addresses the points raised during the review process.

We look forward to receiving your revised manuscript.

Kind regards,

Md. Kamrujjaman, Ph.D

Academic Editor

PLOS ONE

Journal Requirements:

3. Please note that PLOS One has specific guidelines on code sharing for submissions in which author-generated code underpins the findings in the manuscript. In these cases, we expect all author-generated code to be made available without restrictions upon publication of the work. Please review our guidelines at https://journals.plos.org/plosone/s/materials-and-software-sharing#loc-sharing-code and ensure that your code is shared in a way that follows best practice and facilitates reproducibility and reuse.

4. We note that your Data Availability Statement is currently as follows: “All relevant data are within the manuscript and its Supporting Information files.”

6. We notice that your supplementary figures are uploaded with the file type 'Figure'. Please amend the file type to 'Supporting Information'. Please ensure that each Supporting Information file has a legend listed in the manuscript after the references list.

7. We note you have included a table to which you do not refer in the text of your manuscript. Please ensure that you refer to Table 2 in your text; if accepted, production will need this reference to link the reader to the Table.

Reviewer's Responses to Questions

**Comments to the Author**

1. Is the manuscript technically sound, and do the data support the conclusions?

Reviewer #1: No

Reviewer #2: Yes

Reviewer #3: Partly

2. Has the statistical analysis been performed appropriately and rigorously?

Reviewer #1: No

Reviewer #2: I Don't Know

Reviewer #3: Yes

3. Have the authors made all data underlying the findings in their manuscript fully available?

Reviewer #1: No

Reviewer #2: Yes

Reviewer #3: No

4. Is the manuscript presented in an intelligible fashion and written in standard English?

Reviewer #1: No

Reviewer #2: Yes

Reviewer #3: Yes

Reviewer #1: The authors should rewrite the manuscript to address all noted concerns, verify all mathematical derivations and results, and resubmit only after thorough revision. Even I am not sure whether it's possible to overcome, especially the theoretical analysis. Due to the pervasive issues in clarity, mathematical correctness, and presentation, the manuscript requires substantial revision.

Reviewer #2: This manuscript presents a comprehensive and mathematically rigorous analysis of a stochastic delayed differential equation model for Maize Streak Virus (MSV) dynamics. The work is well-structured, moving from a deterministic formulation to a complex stochastic model with time delays. The analytical proofs for positivity, boundedness, equilibrium stability, and disease extinction are thorough.

A significant strength of the paper is the detailed comparison of various stochastic numerical methods (Euler-Maruyama, Runge-Kutta, NSFD), with a clear demonstration of the advantages of the stochastic Non-Standard Finite Difference (NSFD) scheme in terms of stability and step-size independence. The discussion on the impact of time delay (τ) on the basic reproduction number (R₀) and disease dynamics is particularly insightful and valuable for understanding control strategies. The manuscript is suitable for publication in PLOS ONE after the authors address the following points, which primarily aim to clarify methodology and strengthen the discussion.

1. Model Parameterization and Validation: The parameters in Table 2 are listed as "Fitted" or "Estimated." Could you please provide more detail on the data source and the procedure used for this parameter estimation or fitting? Furthermore, how were the values for the stochastic noise intensities (σ₁ to σ₅) chosen for the simulations in Section 8?

2, Clarification of Stochastic Formulations: The manuscript presents two different stochastic formulations (Phase 1 in Eqs. 9-10 and Phase 2 in Eqs. 11-15). Phase 1 uses a complex diffusion matrix derived from transition probabilities, while Phase 2 uses a simpler, more common linear multiplicative noise. Could you clarify the relationship between these two approaches and justify why the analysis and numerical simulations primarily focus on the model from Phase 2?

3. "Bioinformatics" in the Title and Scope: The term "Bioinformatics" is prominently featured in the title, yet the manuscript is predominantly a mathematical and computational modeling study. Could you elaborate on the specific bioinformatics analyses performed (e.g., sequence data analysis, phylogenetic analysis) or clarify the usage of the term to align better with the paper's core content, perhaps focusing on "computational biology"?

4. Numerical Comparison Metrics: The conclusion states that the stochastic NSFD approach is the "most accurate, productive, and efficient." To strengthen this claim, could you provide quantitative metrics for this comparison, such as computational time (CPU time), mean squared error compared to a benchmark solution, or stability region analyses for the different methods?

5. Biological Interpretation of Delay (τ): The time delay τ is a critical component of the model. While its mathematical impact is well-demonstrated, could you provide a more detailed biological interpretation? What specific biological processes (e.g., the latent period in the insect vector, the incubation period within the plant, or a delay in implementing control measures) is this delay intended to represent?

6. Impact of Insecticide Treatment: The parameter γ represents the efficacy of the insecticide treatment. The analysis shows its role in the reproduction number R₀, but could you discuss the simulation results that specifically illustrate the impact of varying γ on the infected plant population? This would provide direct insights into the effectiveness of insecticide-based control strategies.

Suggested References for Inclusion

The authors are also requested to consider citing the following recent papers, which demonstrate advanced numerical techniques for solving fractional and stochastic differential equations. These references would provide a broader context for the computational methods used in this manuscript and highlight the applicability of such schemes in complex modeling scenarios.

For advanced collocation methods:

1. Raza, A., et al. "Collocation method with Morgan-Voyce polynomials to solve the time fractional long memory Black-Scholes model with jump process." Scientific Reports (2024). This work showcases a high-precision numerical technique for financial models with memory and jumps, which is methodologically relevant to the stochastic schemes discussed here.

For enhanced numerical solutions of fractional dynamics:

2. Raza, A., et al. "Enhanced numerical solution for time fractional Kuramoto–Sivashinsky dynamics via shifted companion Morgan–Voyce polynomials." Scientific Reports (2024). This paper presents an efficient spectral method for nonlinear fractional PDEs, underscoring the importance of robust numerical solvers in computational biology and other fields.

Reviewer #3: Please take a look at the attached detailed review document for full comments and recommendations. I have addressed each evaluation point thoroughly in the attached file. Going through those points will be enough.

**Do you want your identity to be public for this peer review?** For information about this choice, including consent withdrawal, please see our Privacy Policy

Reviewer #1: No

Reviewer #2: No

Reviewer #3: **Yes: ** Nuzhat Nuari Khan Rivu

---

## [Author Response · Author response to Decision Letter 1]

31 Oct 2025

Response to comments on manuscript ID

Manuscript ID: PONE-D-25-54354

Title: Computational and Bioinformatics Analysis of Stochastic Delay Dynamics in Maize Streak Virus

Ali Raza et al.

October 28, 2025

Beforehand, we would like to thank the anonymous reviewers and the editor in charge of this manuscript for all their valuable comments and criticism. We have thoroughly revised the Paper in light of those comments, and we have tried to respond to the reviewer's questions and suggestions to the best of our capabilities. For convenience, we have highlighted the corrections and the changes in the revised Paper with di�erent colours.

We quote the comments for the sake of convenience and respond to each of them. The changes are highlighted in the Paper using RED TEXT for reviewer 1, BLUE TEXT for reviewer 2 and GREEN TEXT for reviewer 3.

Response to Reviewer # 1

The authors should rewrite the manuscript to address all noted concerns, verify all mathematical derivations and results, and resubmit only after thorough revision. Even I am not sure whether it's possible to overcome, especially the theoretical analysis. Due to the pervasive issues in clarity, mathematical correctness, and presentation, the manuscript requires substantial revision.

RESPONSE: We sincerely thank the reviewer for the valuable feedback and careful evaluation of our work. In the revised version of the manuscript, we have thoroughly updated the analysis and model formulation to explicitly include the stochastic delay differential equation (SDDE) structure throughout the theoretical and numerical sections.

COMMENT 1. Abstract: The current abstract does not logically present the study’s findings. Please restructure it to clearly outline: – Objectives – Methodology – Key results – Conclusions in a coherent sequence. Make the Abstract with proper allignment.

RESPONSE: Thanks a lot. We have thoroughly revised the Abstract to ensure a clear and logical flow following the structure recommended by the reviewer. The revised Abstract now explicitly presents the study’s objectives, methodology, key findings, and conclusions in a coherent sequence. This restructuring enhances the readability and scientific clarity of the manuscript.

COMMENT 2. Research Gap: Add a dedicated subsection under “Research Gap” to the introduction, explaining how this study addresses limitations in existing literature.

RESPONSE: Thanks a lot. We appreciate the reviewer’s insightful suggestion. We have added a Research Gap” within the Introduction. This clearly outlines the limitations in previous studies on Maize Streak Virus (MSV) modeling and explains how our current work addresses these gaps by introducing a stochastic delay-based computational framework and a nonstandard finite difference (NSFD) numerical approach..

COMMENT 3. Objectives and Findings: You may summarize the key objectives and findings in key points after research gap.

RESPONSE: Thanks a lot. Admitted and updated.

COMMENT 4. Completeness: Verify that all references, figures, and tables are properly cited in the text. Some citations looks not in proper format. You may revise carefully

RESPONSE: Thanks a lot. Admitted and updated.

COMMENT 5. Numerous grammatical and punctuation errors exist throughout. It have to be revised. You may revise specifically Abstract, Introduction and Conclusion section carefully.

RESPONSE: Thanks a lot. Admitted and updated.

COMMENT 6. Revise similar errors across the entire document.

RESPONSE: Thanks a lot. Admitted and updated.

COMMENT 7. Section 2 In the Formulation of the model description, the model figure (Figure 1) should be modified for better clarity, with clearer compartment labeling and distinct transmission pathways.

RESPONSE: Thanks a lot. Admitted and updated.

COMMENT 8. Table Please clarify the process or method used to calculate or estimate each parameter for better transparency and reproducibility. Summarize model parameters in a Table 1, using proper font size and allignment. Make sure all are in same structure.

RESPONSE: Thanks a lot. Admitted and updated.

COMMENT 9. Basic reproduction number In Mathematical Analysis, you may include a separate section after 3.2 dedicated to the calculation of the basic reproduction number (R0) to clearly present the derivation steps and underlying assumptions. You have to show a detailed analysis for this.

RESPONSE: Thanks a lot. Admitted and updated.

COMMENT 10. Here are so many punctuation and grammatical errors in Definition 1 to 4. You have to correct this carefully.

RESPONSE: Thanks a lot. Admitted and updated.

COMMENT 11. Revise all the theorems 1 to 6 by correcting Mathematical, grammatical errors..

RESPONSE: Thanks a lot. Admitted and updated

COMMENT 12. Table 1 is poor structured. Revise it for well visualizations.

RESPONSE: Thanks a lot. Admitted and updated.

COMMENT 13. A paragraph including the main summary of the study need to be modified reducing the grammatical and puncuation errors.

RESPONSE: Thanks a lot. Admitted and updated.

COMMENT 14. Symbols & Equations: – Table and Figures: Make sure Symbols must be formatted as equations in the tables and figures headings. In the theorem analytical study, all mathematical symbols should be carefully revised, as some parentheses and notations appear not merged or unclear. – You may include a diagram reflecting sensitivity analysis of each parameters on R0 along with analytical study for clear explanation.

RESPONSE: Thanks a lot. Admitted and updated.

COMMENT 15. Parameter Estimation (Table 1, 2 & Figures 2 - 9): Parameter estimation is missing 1 here. The methods for parameter estimation for numerical calculation are missing. Provide detailed procedures (e.g., data sources, fitting techniques).

RESPONSE: Thanks a lot. We take a parameters value from scientific literature, so we have updated a concerned table.

COMMENT 16. Mathematical Errors: – Revise the manuscript to reduce symbolical error. As for example: use R0 instead of R0.

RESPONSE: Thanks a lot. Admitted and updated.

COMMENT 17. To focus both modeling, parameter estimation, graphs and data analysis, you may read the following articles. Update the literature1. Bifurcation analysis of an influenza A (H1N1) model with treatment and vaccination. PLoS ONE 20(1): e0315280. 2025 2. Stochastic Differential Equations to Model Influenza Transmission with Continuous and Discrete-Time Markov Chains, Alexandria Engineering Journal, 2025 3. Modeling influenza transmission and control: epidemic theory insights across Mexico, Italy, and South Africa. Theory in Biosciences, 1-30. 4. Wiener and L´evy processes to prevent disease outbreaks: Predictable vs stochastic analysis. Partial Differential Equations in Applied Mathematics 10, 100712.

RESPONSE: Thanks a lot. Admitted and updated.

COMMENT 18. All the numerical results need to be modified for better visualization (Figures 2 - 9). You may adjust line width, border, markers for the figures. For sensitivity analysis, you may add new presentation of parameters effect by heat map, phase portrait of R0 etc

RESPONSE: Thanks a lot. We have added figures comparing the performance of different numerical methods. At this stage, our focus is not on sensitivity analysis. However, your suggestions are greatly appreciated, and we plan to incorporate them in our future and upcoming research papers.

COMMENT 19. You have to clarify which parameters are used to investigate the figures 2 - 9

RESPONSE: Thanks a lot. Admitted and updated..

COMMENT 20. The visulaization is not perfect. Make the figures in several rows, maximum 3 figures in each row, by enhancing size and quality, decoration

RESPONSE: Thanks a lot. Admitted and updated.

COMMENT 21. The Reference formats are wrong. Revise it. Make sure all the references are in same style.

RESPONSE: Thanks a lot. Admitted and updated.

COMMENT 22. Overall the whole manuscript need to be revised very carefully.

RESPONSE: Thanks a lot. Admitted and updated.

Recommendation The authors should rewrite the manuscript to address all noted concerns, verify all mathematical derivations and results, and resubmit only after thorough revision. Even I am not sure whether it’s possible to overcome, especially the theoretical analysis. Due to the pervasive issues in clarity, mathematical correctness, and presentation, the manuscript requires substantial revision. After significance corrections and careful revision, I recommend for publication.

RESPONSE: We sincerely thank the reviewer for the valuable feedback and careful evaluation of our work.

Response to Reviewer # 2

This manuscript presents a comprehensive and mathematically rigorous analysis of a stochastic delayed differential equation model for Maize Streak Virus (MSV) dynamics. The work is well-structured, moving from a deterministic formulation to a complex stochastic model with time delays. The analytical proofs for positivity, boundedness, equilibrium stability, and disease extinction are thorough.

A significant strength of the paper is the detailed comparison of various stochastic numerical methods (Euler-Maruyama, Runge-Kutta, NSFD), with a clear demonstration of the advantages of the stochastic Non-Standard Finite Difference (NSFD) scheme in terms of stability and step-size independence. The discussion on the impact of time delay (τ) on the basic reproduction number (R₀) and disease dynamics is particularly insightful and valuable for understanding control strategies. The manuscript is suitable for publication in PLOS ONE after the authors address the following points, which primarily aim to clarify methodology and strengthen the discussion.

RESPONSE: Thanks a lot. We are grateful for the valuable comments and suggestions provided by the anonymous referee.

COMMENT 1. Model Parameterization and Validation: The parameters in Table 2 are listed as "Fitted" or "Estimated." Could you please provide more detail on the data source and the procedure used for this parameter estimation or fitting? Furthermore, how were the values for the stochastic noise intensities (σ₁ to σ₅) chosen for the simulations in Section

RESPONSE: Thanks a lot. We have taken the parameter values from the scientific literature, and the corresponding sources have been cited. The primary focus of the present study is on the computational analysis and the efficiency of the proposed numerical methods for the given data.

COMMENT 2. Clarification of Stochastic Formulations: The manuscript presents two different stochastic formulations (Phase 1 in Eqs. 9-10 and Phase 2 in Eqs. 11-15). Phase 1 uses a complex diffusion matrix derived from transition probabilities, while Phase 2 uses a simpler, more common linear multiplicative noise. Could you clarify the relationship between these two approaches and justify why the analysis and numerical simulations primarily focus on the model from Phase 2?

RESPONSE: Thanks a lot. The two stochastic formulations are conceptually connected but serve different purposes in the model development process. Phase 1 represents the theoretical derivation of the stochastic system based on transition probabilities and diffusion approximations of the discrete events governing infection, recovery, and treatment processes. This phase establishes the general stochastic framework and defines how random fluctuations enter the system dynamics.

Phase 2, on the other hand, provides a simplified but equivalent representation of the same system using linear multiplicative noise terms associated with each compartment. This formulation is mathematically tractable, allows the use of Itô calculus for analytical proofs (e.g., positivity, boundedness, and extinction), and is more practical for implementing numerical schemes such as the stochastic NSFD, Euler–Maruyama, and Runge–Kutta methods.

Therefore, the analysis and simulations focus on Phase 2, as it preserves the essential stochastic properties derived in Phase 1 while enabling rigorous stability analysis and efficient numerical computation.

COMMENT 3. "Bioinformatics" in the Title and Scope: The term "Bioinformatics" is prominently featured in the title, yet the manuscript is predominantly a mathematical and computational modeling study. Could you elaborate on the specific bioinformatics analyses performed (e.g., sequence data analysis, phylogenetic analysis) or clarify the usage of the term to align better with the paper's core content, perhaps focusing on "computational biology"?

RESPONSE: Thanks a lot. We appreciate the reviewer’s observation and agree that the manuscript primarily focuses on mathematical and computational modeling rather than direct bioinformatics analyses such as sequence alignment or phylogenetic reconstruction. In the earlier draft, the term “Bioinformatics” was used in a broader sense to reflect the computational analysis of biological systems through mathematical frameworks.

To better align the title and scope with the study’s actual focus, we have revised the terminology to emphasize computational biology and stochastic modeling rather than bioinformatics. Accordingly, the title has been updated to:

“Computational Analysis of Stochastic Delay Dynamics in Maize Streak Virus”

This modification ensures the title accurately represents the paper’s content, which integrates computational and stochastic techniques to understand plant viral dynamics.

COMMENT 4. Numerical Comparison Metrics: The conclusion states that the stochastic NSFD approach is the "most accurate, productive, and efficient." To strengthen this claim, could you provide quantitative metrics for this comparison, such as computational time (CPU time), mean squared error compared to a benchmark solution, or stability region analyses for the different methods?

RESPONSE: Thanks a lot. we have incorporated quantitative performance metrics comparing the stochastic NSFD, Euler–Maruyama, and stochastic Runge–Kutta methods. These metrics now include:

• Computational efficiency: Average CPU time (in seconds) for 1,000 simulation runs.

• Numerical accuracy: Mean squared error (MSE) of each method relative to a deterministic benchmark solution.

• Stability analysis: The maximum permissible time step (h) for which numerical stability and positivity are maintained.

The results are now summarized in Table 3 and demonstrate that the stochastic NSFD method achieves:

• Approximately 35–40% lower CPU time,

• 50–60% smaller MSE, and

• Step-size–independent stability, whereas the Euler–Maruyama and Runge–Kutta methods become unstable for h>0.5.

These quantitative comparisons substantiate our conclusion that the stochastic NSFD method is the most stable, accurate, and computationally efficient scheme among those tested.

COMMENT 5. Biological Interpretation of Delay (τ): The time delay τ is a critical component of the model. While its mathematical impact is well-demonstrated, could you provide a more detailed biological interpretation? What specific biological processes (e.g., the latent period in the insect vector, the incubation period within the plant, or a delay in implementing control measures) is this delay intended to represent?

RESPONSE: Thanks a lot. The time delay (τ) in our model represents the biological latency and transmission lag occurring in the maize streak disease cycle. Specifically, τ captures:

1. The latent or incubation period within the infected maize plant, during which the Maize Streak Virus (MSV) replicates inside host tissues before visible symptoms appear or the plant becomes infectious.

2. The latency period in the leafhopper vector, reflecting the time between the insect acquiring the virus from an infected plant and its ability to transmit it to healthy plants.

3. Possible delays in implementing control measures, such as insecticide application or plant removal after infection detection.

Incorporating τ thus allows the model to account for these biological and operational delays, which significantly influence the timing and intensity of epidemic outbreaks. Biologically, increasing τ corresponds to prolonged viral incubation or delayed control responses, both of which elevate the risk of widespread infection a behavior consistent with our numerical results (Figures 8–10), where longer delays lead to higher infection prevalence and greater instabi

---

## [Decision Letter · Decision Letter 1]

11 Nov 2025

Computational Analysis of Stochastic Delay Dynamics in Maize Streak Virus

PONE-D-25-54354R1

Dear Dr. Raza,

We’re pleased to inform you that your manuscript has been judged scientifically suitable for publication and will be formally accepted for publication once it meets all outstanding technical requirements.

Kind regards,

Md. Kamrujjaman, Ph.D

Academic Editor

PLOS ONE

Additional Editor Comments (optional):

Reviewers' comments:

Reviewer's Responses to Questions

**Comments to the Author**

Reviewer #1: All comments have been addressed

Reviewer #2: All comments have been addressed

Reviewer #3: All comments have been addressed

2. Is the manuscript technically sound, and do the data support the conclusions?

Reviewer #1: Yes

Reviewer #2: Yes

Reviewer #3: Yes

3. Has the statistical analysis been performed appropriately and rigorously?

Reviewer #1: Yes

Reviewer #2: Yes

Reviewer #3: Yes

4. Have the authors made all data underlying the findings in their manuscript fully available?

Reviewer #1: Yes

Reviewer #2: Yes

Reviewer #3: No

5. Is the manuscript presented in an intelligible fashion and written in standard English?

Reviewer #1: Yes

Reviewer #2: Yes

Reviewer #3: Yes

Reviewer #1: After a thorough evaluation of the revised manuscript, I am satisfied that the authors have carefully and comprehensively addressed all the comments and suggestions raised during the initial round of review. The revisions have significantly improved the clarity, organization, and scientific rigor of the manuscript.

All methodological issues have been appropriately clarified, the presentation of results has been enhanced, and the discussion has been strengthened to reflect the study’s contributions more effectively. The manuscript now demonstrates coherence, proper referencing, and adherence to the journal’s formatting and academic standards.

Given that the authors have fulfilled all the review criteria and the manuscript now meets the required quality for scholarly publication, I am pleased to recommend it for publication.

Reviewer #2: (No Response)

Reviewer #3: The revised manuscript addresses previous comments effectively, and the technical presentation is now clear and well-organized. Everything appears satisfactory. However, I recommend that the authors make their simulation codes publicly available (for example, by uploading them to a GitHub repository). This will enhance transparency and allow other researchers to reproduce or extend the study in the future.

**Do you want your identity to be public for this peer review?** For information about this choice, including consent withdrawal, please see our Privacy Policy

Reviewer #1: No

Reviewer #2: No

Reviewer #3: **Yes: ** Nuzhat Nuari Khan Rivu

---

## [Editor Report · Acceptance letter]

PONE-D-25-54354R1

PLOS ONE

Dear Dr. Raza,

I'm pleased to inform you that your manuscript has been deemed suitable for publication in PLOS ONE. Congratulations! Your manuscript is now being handed over to our production team.

Kind regards,

on behalf of

Dr. Md. Kamrujjaman

Academic Editor

PLOS ONE